# MAP: Low-compute Model Merging with Amortized Pareto Fronts via Quadratic Approximation

**Lu Li[1*], Tianyu Zhang[2,3*], Zhiqi Bu[4*], Suyuchen Wang [5], Huan He[1],**
**Jie Fu[6], Jiang Bian[7], Yonghui Wu[7], Yong Chen[1], Yoshua Bengio[2]**

[1] University of Pennsylvania  [2] MILA  [3] ServiceNow  [4] Amazon AGI  [5] Université de Montréal
[6] HKUST  [7] University of Florida
`luli1@sas.upenn.edu, {Huan.He, ychen123}@pennmedicine.upenn.edu,`
`{tianyu.zhang, yoshua.bengio}@mila.quebec,`
`woodyx218@gmail.com, suyuchen.wang@umontreal.ca, fujie@ust.hk,`
`{bianjiang, yonghui.wu}@ufl.edu`

## ABSTRACT

Model merging has emerged as an effective approach to combining multiple single-task models into a multitask model. This process typically involves computing a weighted average of the model parameters without additional training. Existing model-merging methods focus on improving average task accuracy. However, interference and conflicts between the objectives of different tasks can lead to trade-offs during the merging process. In real-world applications, a set of solutions with various trade-offs can be more informative, helping practitioners make decisions based on diverse preferences. In this paper, we introduce a novel and low-compute algorithm, **Model Merging with Amortized Pareto Front (MAP)**. MAP efficiently identifies a Pareto set of scaling coefficients for merging multiple models, reflecting the trade-offs involved. It amortizes the substantial computational cost of evaluations needed to estimate the Pareto front by using quadratic approximation surrogate models derived from a preselected set of scaling coefficients. Experimental results on vision and natural language processing tasks demonstrate that MAP can accurately identify the Pareto front, providing practitioners with flexible solutions to balance competing task objectives. We also introduce Bayesian MAP for scenarios with a relatively low number of tasks and Nested MAP for situations with a high number of tasks, further reducing the computational cost of evaluation.

## 1 INTRODUCTION

Large pre-trained foundation models have become widely available for many real-world applications (Wornow et al., 2023; Thirunavukarasu et al., 2023; Cui et al., 2024). This increasing availability has led to a popular practice of fine-tuning these models to adapt them to a wide range of downstream tasks. Practitioners can independently fine-tune the same pre-trained model, such as CLIP-style models (Radford et al., 2021; Wu et al., 2023; Zhai et al., 2023), large language models (Brown et al., 2020; Rozière et al., 2023; Touvron et al., 2023; Jiang et al., 2024), etc., and then release the fine-tuned models without disclosing the training data. As the deployment of such fine-tuned models increases, combining models with identical architectures and initializations has emerged as a promising approach to combine their respective capabilities. This is useful, especially in scenarios where the training data for each task is private and cannot be shared, such as individual-level patient data in hospitals and behavioral data in social media recommendation systems.

Existing methods for merging models typically involve calculating a weighted average of the parameters from multiple models to enhance performance uniformly across various tasks. However, this approach often overlooks the conflicts among the diverse objectives of these tasks, which can

---

[*]Equal contribution.

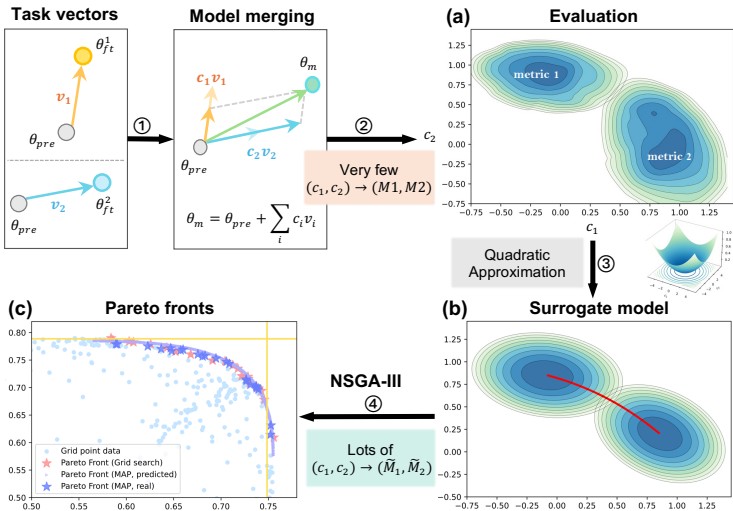

Figure 1: Illustration of the overall process of MAP for the case of two tasks. **Step 1**: Select 2 tasks and compute their corresponding task vectors. **Step 2**: Sample a few scaling coefficients $\mathbf{c}$ and query the evaluation metrics for each task, respectively. **Step 3**: Use the quadratic model as a surrogate model to approximate the mapping $\mathbf{c} \rightarrow$ *metrics*. **Step 4**: Use the NSGA-III algorithm with the surrogate objective functions to find amortized Pareto fronts. **(a)**: Contour plot of the actual accuracy landscape for the ViT-B/32 model (Dosovitskiy et al., 2020) obtained from 100 scaling coefficients (sampled uniformly) evaluated on the SUN397 (Xiao et al., 2016) and Cars (Krause et al., 2013) datasets. **(b)**: Contour plot of the fitted quadratic functions. Red lines represent the Pareto front in the decision variable $(c_1, c_2)$ space. **(c)**: Example of the resulting Pareto fronts. The Pareto front (Grid search) is regarded as the ground truth given the sufficient number of grid points evaluated. The Pareto front (MAP, predicted) is the amortized Pareto front. The Pareto front (MAP, real) corresponds to the same $\{(c_1, c_2)\}$ but is re-evaluated to obtain the ground truth metrics for comparison. The yellow lines indicate the evaluated performance of the fine-tuned single-task models.

lead to trade-offs in model performance across different tasks. In real-world applications, it is often more beneficial to obtain a set of Pareto optimal solutions rather than a single model. Such solutions allow practitioners to choose among different trade-offs based on their specific needs. For example, hospitals specializing in certain areas might prefer a model that excels in tasks relevant to their specialty while maintaining adequate performance across a broader spectrum of diseases.

In this paper, we introduce a novel method that identifies the Pareto front without retraining the models to be merged. Our algorithm utilizes a quadratic approximation of the evaluation metric. Furthermore, we enhance it with a Bayesian adaptive sampling method and a nested merging scheme, which further reduces the computational cost. We validate our method across a diverse set of tasks, spanning from vision to natural language processing, and demonstrate its applicability to a variety of architectures, including ResNets (He et al., 2016), ViT (Dosovitskiy et al., 2020), and large language models (Brown et al., 2020; Rozière et al., 2023; Touvron et al., 2023; Jiang et al., 2024). Our results confirm that this novel approach supports the seamless integration of diverse model capabilities and aligns more closely with various real-world preferences by providing a set of optimal fronts across the tasks.

**Contributions**  The main contributions of this paper are:

**C1** We propose the MAP algorithm, which utilizes quadratic surrogate models to approximate evaluation metric functions, thereby amortizing the computation of Pareto fronts.

**C2** We demonstrate the effectiveness of MAP across a diverse set of architectures, including vision models (e.g., ResNet, ViT) and large language models, highlighting its generalizability and flexibility across different domains.

**C3** We introduce two variants of MAP: the nested-merging MAP, which reduces computational complexity from $O(N \cdot 2^N)$ to $O(N \log N)$, and the Bayesian MAP, which efficiently queries computationally expensive evaluations based on loss information.

**C4** To the best of our knowledge, this paper is the first work to estimates the Pareto front for task-vector-based model-merging methods with low compute, and without relying on gradient descent for deep neural networks. Our code is available at `https://github.com/luli-git/MAP`.

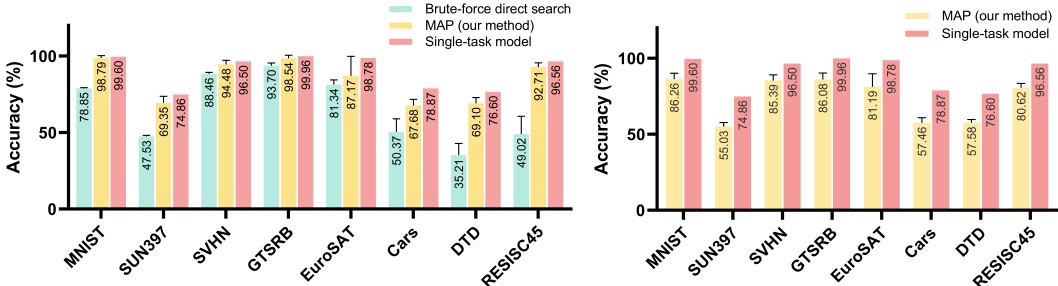

Figure 2: **Left**: Comparison of our method with direct search for merged ViT-B/32 models (Dosovitskiy et al., 2020), based on evaluation results of 250 combinations of scaling coefficients. Our method identifies a more diverse set of solutions across eight tasks within the same computational budget. Both methods aim to maximize the performance of one task while ensuring that all other tasks meet a minimum threshold of $40\%$. The bar plot displays the maximized accuracy for each task. **Right**: When the threshold is increased to $65\%$ of the single-task model's performance, the brute-force direct search method fails to find any feasible solutions within the same computational budget.

In Figure 2, we compare our method to direct search. It is important to note that evaluating models in our problem setting is computationally expensive; as a result, our proposed method can only query the evaluation pipeline a limited number of times. We choose to compare with direct brute-force search within the scaling coefficient spaces. When the number of tasks is low, the Pareto fronts obtained through direct brute-force search can be considered ground truth because the scaling coefficient spaces have been sufficiently explored. However, when the number of tasks is high, we may not be able to derive reasonable ground truth Pareto fronts using direct brute-force search. Nonetheless, we can still demonstrate the efficiency of our method by comparing the winning ratio of MAP to that of direct brute-force search.

## 2 BACKGROUND

### 2.1 MODEL MERGING

*Model merging* aims to combine two or more trained models into a single model to leverage the strengths of each and improve overall performance (Deep et al., 2024; Shoemake, 1985; Wang et al., 2024b). A recent work by Ilharco et al. (2022) introduced *task arithmetic* as a simple and effective method for model merging. The task vector for task $n$ is defined as $\mathbf{v}_n = \boldsymbol{\theta}_{ft}^n - \boldsymbol{\theta}_{pre}$, which is the element-wise difference between the pre-trained weights and the fine-tuned weights for task $n$. To perform model merging with task vectors, we can compute the merged model as $\boldsymbol{\theta}_m = \boldsymbol{\theta}_{pre} + \sum_{n=1}^{N} c_n \mathbf{v}_n$, where $c_n$ are the scaling coefficients that have been shown to be essential for the performance of the merged model (Yadav et al., 2024; Yang et al., 2023).

Denoting the metric of task $n$ as $M_n$, most existing approaches to model merging aim to improve an equal weight average metric $\frac{1}{N} \sum_{n=1}^{N} M_n$. This target implies that the user of the algorithm has no preferences between tasks. However, in real-world applications, users may have biased preferences for the importance of tasks, necessitating trade-offs in model performance. In such cases, the goal of model merging shifts from the simple average metric to addressing a broader question:

*Given any preferences over the set of tasks, what is the best way to merge the models?*

## 2.2 PARETO FRONTS

**Definition 1** (Pareto dominance). *Let $X$ be a set representing the solution space, where each element $x \in X$ is a possible solution to the multi-objective optimization problem. Let there be $n$ objectives, and define an evaluation loss function $f_i : X \to \mathbb{R}$, where $i \in \{1, 2, \ldots, n\}$.*

*Given two solutions $x, y \in X$, we define that $x$ Pareto dominates $y$, denoted by $x \succ_P y$, if and only if:*

$$\forall i \in \{1, 2, \ldots, n\}, f_i(x) \le f_i(y) \text{ and } \exists j \in \{1, 2, \ldots, n\}, f_j(x) < f_j(y)$$

**Definition 2** (Pareto optimal solutions). *The Pareto front is the set of solutions in the solution space $X$ that are not Pareto dominated by any other solutions in $X$. The Pareto front is given by:*

$$PF = \{x \in X \mid \nexists y \in X \text{ s.t. } y \succ_P x\} \tag{1}$$

Pareto optimal solutions have been studied in multi-task (multi-objective) learning (MTL) (Sener & Koltun, 2018; Lin et al., 2019). However, in most of the studies, approximating the Pareto front is computationally expensive and data inefficient. We introduce our method, MAP, a computationally efficient method to find the Pareto front for model merging.

## 3 METHODS

In this section, we discuss the proposed MAP method in detail. To enhance readability, we present a table of notations in Table 5.

### 3.1 QUADRATIC APPROXIMATION OF EVALUATION METRIC

Given the task vectors $\{\mathbf{v}_n\}_{\{n=1,\ldots,N\}}$ and the initialization $\boldsymbol{\theta}_{\text{pre}} \in \mathbb{R}^d$, we denote the merged model parameters as $\boldsymbol{\theta}_m(\mathbf{c}) = \boldsymbol{\theta}_{\text{pre}} + \mathbf{V}\mathbf{c} = \boldsymbol{\theta}_{\text{pre}} + \sum_{n=1}^{N} c_n \mathbf{v}_n$, where $\mathbf{V} = \text{concat}(\mathbf{v}_1, \ldots, \mathbf{v}_N) \in \mathbb{R}^{d \times N}$ is the task matrix and $\mathbf{c} = [c_1, \ldots, c_N]^\top \in \mathbb{R}^N$ is the vector of scaling coefficients for the task vectors.

Let $M_n(\mathbf{c}) = M_n(\theta_m(\mathbf{c}))$ denote the evaluation metric for task $n$ of the merged model. We aim to optimize the evaluation metric for each task via the multi-objective optimization problem (MOOP)[1]:

$$\min_{c_1,\ldots,c_N} M_1(\mathbf{c}), \ldots, M_N(\mathbf{c}) \tag{2}$$

This problem has $N$ variables and $N$ objectives, and we seek the Pareto optimal solutions.

Motivated by the observation that the finetuned models tend to have parameters close to the pretrained model (as shown in Table 1), we present the following assumption.

**Assumption 1.** *The task vectors $\mathbf{v}_n$ have small norms relative to the pretrained model parameters $\boldsymbol{\theta}_{\text{pre}}$, i.e., $|\mathbf{v}_n| \ll |\boldsymbol{\theta}_{\text{pre}}|$ for $n = 1, \ldots, N$. Additionally, the evaluation metrics $M_n(\theta)$ are sufficiently smooth around $\boldsymbol{\theta}_{\text{pre}}$, such that higher-order terms in their Taylor expansions are negligible within the region defined by the task vectors.*

Denote $p$ as the number of parameters in the pre-trained model and also as the number of parameters in each task vector. Let $N$ represent the number of tasks.

To derive our algorithm, MAP, we utilize the second-order Taylor expansion to approximate $M_n$:

$$M_n(\mathbf{c}) \equiv M_n(\boldsymbol{\theta}_m(\mathbf{c})) = M_n(\boldsymbol{\theta}_{\text{pre}}) + \nabla M_n(\boldsymbol{\theta}_{\text{pre}})^\top (\boldsymbol{\theta}_m(\mathbf{c}) - \boldsymbol{\theta}_{\text{pre}})$$

$$+ \frac{1}{2}(\boldsymbol{\theta}_m(\mathbf{c}) - \boldsymbol{\theta}_{\text{pre}})^\top \mathbf{H}_n(\boldsymbol{\theta}_{\text{pre}})(\boldsymbol{\theta}_m(\mathbf{c}) - \boldsymbol{\theta}_{\text{pre}}) + R_n(\boldsymbol{\theta}_m(\mathbf{c}) - \boldsymbol{\theta}_{\text{pre}})$$

$$= \underbrace{M_n(\boldsymbol{\theta}_{\text{pre}})}_{\in \mathbb{R}} + \underbrace{\nabla M_n(\boldsymbol{\theta}_{\text{pre}})^\top}_{\in \mathbb{R}^{1 \times p}} \underbrace{\mathbf{V}}_{\in \mathbb{R}^{p \times N}} \underbrace{\mathbf{c}}_{\in \mathbb{R}^{N \times 1}} + \frac{1}{2} \underbrace{(\mathbf{V}\mathbf{c})^\top}_{\in \mathbb{R}^{1 \times p}} \underbrace{\mathbf{H}_n(\boldsymbol{\theta}_{\text{pre}})}_{\in \mathbb{R}^{p \times p}} \underbrace{\mathbf{V}\mathbf{c}}_{\in \mathbb{R}^{p \times 1}} + \underbrace{R_n}_{\in \mathbb{R}}$$

where $\mathbf{H}_n(\boldsymbol{\theta}_{\text{pre}}) = \nabla^2 M_n(\boldsymbol{\theta}_{\text{pre}}) \in \mathbb{R}^{d \times d}$ is the Hessian matrix and $R_n(\boldsymbol{\theta}_m(\mathbf{c}) - \boldsymbol{\theta}_{\text{pre}}) = R_n(\mathbf{V}\mathbf{c})$ is the third-order remainder, which, as stated in Assumption 1, is negligible when $||\mathbf{V}\mathbf{c}||^3 = ||\boldsymbol{\theta}_m(\mathbf{c}) -$

---

[1]The evaluation metric $M$ can be differentiable (e.g., mean square loss or cross-entropy/perplexity) or not necessarily (e.g., classification accuracy, F1 score, BLEU, or Rouge)

$\boldsymbol{\theta}_{\text{pre}}||^3$ is small. Note that the second-order Taylor expansion is widely used (McCandlish et al., 2018; Bu et al., 2024; Kaplan et al., 2020; Hoffmann et al., 2022) and provides a close approximation (see Figure 2 (a) and (b) and more discussion in Appendix 5).

Table 1: We compute the $L_1$ norm of the weight matrices of the pretrained models and the task vectors for each of the 8 tasks using the ViT-B/32 model, and compute the ratio.

| Metric | SUN397 | Cars | DTD | SVHN | RESISC45 | MNIST | GTSRB | EuroSAT |
|---|---|---|---|---|---|---|---|---|
| $\|\boldsymbol{\theta}_{pre}\|_1$ | 1,270,487 | 1,270,487 | 1,270,487 | 1,270,487 | 1,270,487 | 1,270,487 | 1,270,487 | 1,270,487 |
| $\|\mathbf{v}_n\|_1$ | 21,055 | 20,127 | 13,621 | 19,349 | 18,409 | 17,578 | 16,712 | 15,941 |
| $\|\mathbf{v}_n\|_1/\|\boldsymbol{\theta}_{pre}\|_1(\%)$ | 1.66% | 1.58% | 1.07% | 1.52% | 1.45% | 1.38% | 1.32% | 1.25% |

To validate the approximation, we calculate the $L_1$ norm of the parameters for the 8 task vectors and their ratios relative to the $L_1$ norm of the pretrained model. As shown in Table 1, the ratio between the $L_1$ norm of the task vectors and the $L_1$ norm of the weight matrices of the pretrained model is approximately $1\%$ to $2\%$ for each of the 8 tasks.

We further examine the density plot of the absolute values of the weight matrices of the 8 task vectors (Figure 3). We find that the maximum magnitude of the task vectors is 0.00859, while the mean is 1.57336 $\times10^{-4}$. These findings motivate and validate our key assumption 1 of using a second-order Taylor expansion to approximate the evaluation metrics.

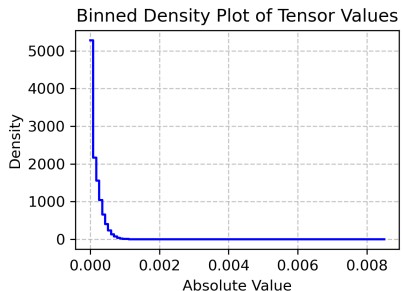

Leveraging this quadratic approximation, we can define surrogate models for each task $N$,

$$\tilde{M}_n(\mathbf{c}; \mathbf{A}_n, \mathbf{b}_n, e_n) \equiv \frac{1}{2}\mathbf{c}^\top \mathbf{A}_n\mathbf{c} + \mathbf{b}_n^\top \mathbf{c} + e_n \quad (3)$$

Figure 3: Density plot of the absolute values of the weight matrices of the 8 task vectors.

and optimize the proxy problem of Equation (2) via

$$\min_{c_1,\ldots,c_N} \tilde{M}_1(\mathbf{c}),\ldots,\tilde{M}_N(\mathbf{c}) \quad (4)$$

where $\mathbf{A}_n \in \mathbb{R}^{N\times N}$ is parameter matrix approximating $\mathbf{V}^\top \mathbf{H}_n(\boldsymbol{\theta}_{\text{pre}})\mathbf{V}$; $\mathbf{b}_n \in \mathbb{R}^N$ is a parameterized vector targeting to $\mathbf{V}^\top \nabla M_n(\boldsymbol{\theta}_{\text{pre}})$; $e_n$ is a parameter scalar which estimates $M_n(\boldsymbol{\theta}_{\text{pre}}) + R_n$.

Importantly, Equation (4) is parameterized in contrast to Equation (2), with a total of $\frac{(N+1)(N+2)}{2}$ surrogate model parameters in $(e_n, \mathbf{b}_n, \mathbf{A}_n)$, i.e., 1 coefficient for $e_n$, $N$ coefficients in $\mathbf{b}_n$ and $N(N+1)/2$ coefficients in $\mathbf{A}_n$ due to its symmetry $(1 + N + \frac{N(N+1)}{2} = \frac{(N+1)(N+2)}{2})$. We do not calculate the gradient or Hessian to obtain the parameters $\mathbf{A}_n, \mathbf{b}_n, e_n$. Instead, we estimate them by minimizing the MSE between $M_n$ and $\tilde{M}_n$, $n = 1,\ldots,N$:

$$\mathbf{A}_n^*, \mathbf{b}_n^*, e_n^* = \arg\min_{\mathbf{A}_n,\mathbf{b}_n,e_n} \sum_{\mathbf{c}\in\Omega} |M_n(\boldsymbol{\theta}_{\text{merge}}(\mathbf{c})) - \tilde{M}_n(\mathbf{c}; \mathbf{A}_n, \mathbf{b}_n, e_n)|^2 \quad (5)$$

where $\Omega = \{\mathbf{c}^{(1)},\ldots,\mathbf{c}^{(K)}\}$ is the set of $\mathbf{c}$ and $M_n(\boldsymbol{\theta}_{\text{merge}}(\mathbf{c}))$ is the corresponding evaluation metric. Below we discuss 3 extensive cases of this problem.

**Case 1:** If the evaluation metric spans the whole real-number axis $\mathbb{R}$, we use the vanilla form of the surrogate model in Equation (3). As shown in Corollary 1, we have a closed-form solution for Equation (5)

**Corollary 1** (Closed-form Solution for Surrogate Model Parameters). *Under Assumption 1, for each task $n = 1,\ldots,N$, the optimization problem 5 is equivalent to solving a linear regression where the predictors include all quadratic, interaction, linear, and constant terms of $\mathbf{c}$. The closed-form solution for the parameters is given by*

$$\begin{pmatrix} vec(A_n^*) \\ \mathbf{b}_n^* \\ e_n^* \end{pmatrix} = \left(\mathbf{C}_n^\top \mathbf{C}_n\right)^{-1} \mathbf{C}_n^\top \mathbf{y}_n$$

where $\mathbf{C}_n(\mathbf{c}) = (c_1^2, c_2^2, \ldots, c_N^2, c_1c_2, c_1c_3, \ldots, c_{N-1}c_N, c_1, c_2, \ldots, c_N, 1)$, and $\mathbf{y}_n$ is the vector of observed metrics $M_n(\theta_m(\mathbf{c}))$ for all $\mathbf{c} \in \Omega$. Please refer to Corollary 3 for a more detailed corollary.

**Case 2:** When the evaluation metric is restricted to a specific range $[l, u]$, e.g. accuracy is restricted to $[0, 1]$, we can apply a sigmoid transformation to the quadratic term, i.e.

$$\tilde{M}_n(\mathbf{c}; \mathbf{A}_n, \mathbf{b}_n, e_n) \equiv (u - l)\sigma(e_n + \mathbf{b}_n^\top \mathbf{c} + \frac{1}{2}\mathbf{c}^\top \mathbf{A}_n \mathbf{c}) + l.$$

**Case 3:** Similarly, when the evaluation metric is restricted to $[l, +\infty)$, applying a softplus transformation ensures non-negativity while allowing smooth growth. In this case, we define:

$$\tilde{M}_n(\mathbf{c}; \mathbf{A}_n, \mathbf{b}_n, e_n) \equiv \text{softplus}(e_n + \mathbf{b}_n^\top \mathbf{c} + \frac{1}{2}\mathbf{c}^\top \mathbf{A}_n \mathbf{c}) + l,$$

where $\text{softplus}(x) = \ln(1 + e^x)$.

In Cases 2 and 3, the parameters $u$ and $l$ are not learnable parameters but predefined based on the feasible range of the metric. The introduction of non-linear functions in $\tilde{M}_n$ eliminates closed-form solutions, necessitating the use of gradient descent to solve the optimization problem 5. Despite this, the approach remains computationally efficient since the number of parameters in the surrogate model is $\frac{(N+1)(N+2)}{2}$. We emphasize that gradient descent on the surrogate model is computationally efficient. Compared to training full-scale deep learning models, this method requires significantly fewer computations. For instance, when merging eight models, the parameter count is only 45, which is much smaller than the millions of parameters typically found in deep learning architectures.

## 3.2 Model merging with amortized Pareto fronts

In this section, we introduce our generalized algorithm for estimating the amortized Pareto fronts. As mentioned in Section 3.1, we approximate the evaluation metric $M_n(\cdot)$ by a surrogate quadratic model $\tilde{M}_n(\cdot)$. We then utilize $\tilde{M}_n(\cdot)$ to compute the amortized Pareto fronts. Please see the detailed experiments in Section 4 and the algorithm details in Algorithm 1. Different from other Pareto multi-task learning algorithms, our algorithm does not require calculating the gradients or performing gradient descent on the deep learning model parameters.

---
**Algorithm 1** MAP
---
**Input:** Pretrained model $\theta_{pre}$, fine-tuned models $\{\theta_{ft}^n\}_{n=1}^N$.
Compute task vectors $\{\mathbf{v}_n = \theta_{ft}^n - \theta_{\text{pre}} \mid n \in 1, \ldots, N\}$.
Sample $K$ vectors of $\mathbf{c} \in \mathbb{R}^N$. Denote the set as $\Omega$.
**for** $n \in [N]$ **do**
    **for** $\mathbf{c} = [c_1, ..., c_N] \in \Omega$ **do**
        Compute $\theta(\mathbf{c}) = \theta_{\text{pre}} + c_1\mathbf{v}_1 + \ldots + c_N\mathbf{v}_N$.
        Obtain the evaluation metric $M_n(\theta(\mathbf{c}))$.
    **end**
    Fit the quadratic approximation surrogate model $\tilde{M}_n$ by learning $\mathbf{A}_n^*, \mathbf{b}_n^*, e_n^*$ in Equation (5).
**end**
Apply MOOP algorithm (e.g. NSGA-III) to $\{\tilde{M}_n\}$ and find the Pareto front

---

## 3.3 Further methods to bring down the computation cost when the number of tasks is high

In addition to MAP, we also introduce nested merging MAP (NMMAP) and Bayesian MAP (BMAP), which are variants of MAP designed to further reduce computation costs.

For a small number of tasks ($N \leq 3$), we use Bayesian adaptive sampling, inspired by Bayesian optimization. Unlike plain MAP (Algorithm 1), which samples a single set of scaling weights $\mathbf{c}$ and evaluates metrics $M_n(\theta(\mathbf{c}))$, Bayesian adaptive sampling iteratively samples $\mathbf{c}$ across multiple rounds, with each round informed by prior evaluations.

The process starts with uniform sampling of scaling coefficients $\{(c_1, \ldots, c_N)\}$, from $[0, 1]^N$, evaluating the merged models $\theta_m(\mathbf{c}_i)$ for tasks 1 to $N$, and calculating the $L_2$ loss. We compute a posterior distribution based on an acquisition function, like the upper confidence bound, for each bin.

We iteratively update surrogate models for each task, and after meeting the stopping criterion, generate $(\mathbf{c}, \{\tilde{M}_n(\theta_m(\mathbf{c}))\}_{n=1}^N)$ samples. A MOOP algorithm (e.g., NSGA-III) is then applied to compute the Pareto front from $\tilde{M}_n$.

As illustrated in Figure 4 (a), BMAP uses Bayesian optimization to determine the distribution of scaling coefficients for task vectors when approximating the surrogate model in MAP. For more details and experiments, please refer to Appendix E.3.

When the number of tasks $N$ is high and computational resources are also limited, NMMAP reduces the number of evaluations from $O(TN2^N)$ to $O(N \log N)$. For example, tasks 1 to 8 cover Cars, GTSRB, DTD, SUN397, RESISC45, MNIST, EuroSAT, and SVHN. With nested merging, in the first round, we merge $(\boldsymbol{\theta}_{ft}^1, \boldsymbol{\theta}_{ft}^2)$ into $\boldsymbol{\theta}_{\text{merge}}^{1,2}$. Similarly, we merge $(\boldsymbol{\theta}_{ft}^3, \boldsymbol{\theta}_{ft}^4)$ into $\boldsymbol{\theta}_{\text{merge}}^{3,4}$, $(\boldsymbol{\theta}_{ft}^5, \boldsymbol{\theta}_{ft}^6)$ into $\boldsymbol{\theta}_{\text{merge}}^{5,6}$ and $(\boldsymbol{\theta}_{ft}^7, \boldsymbol{\theta}_{ft}^8)$ into $\boldsymbol{\theta}_{\text{merge}}^{7,8}$ in parallel. Next, we merge $(\boldsymbol{\theta}_{\text{merge}}^{1,2}, \boldsymbol{\theta}_{\text{merge}}^{3,4})$ into $\boldsymbol{\theta}_{\text{merge}}^{1,2,3,4}$, and $(\boldsymbol{\theta}_{\text{merge}}^{5,6}, \boldsymbol{\theta}_{\text{merge}}^{7,8})$ into $\boldsymbol{\theta}_{\text{merge}}^{5,6,7,8}$ simultaneously and finally into $\boldsymbol{\theta}_{\text{merge}}^{1,2,3,4,5,6,7,8}$. Note that the model merging order is illustrative and chosen to support the exposition.

This approach achieves a 250x speedup when the number of tasks is 8, while also outperforming direct search methods. However, it can no longer output the complete Pareto front for all tasks, and practitioners need to be able to quantify their preferences during the merging since it merges models in pairs in a nested fashion, as shown in Figure 4 (b). For more details and experiments, please refer to Appendix E.2.

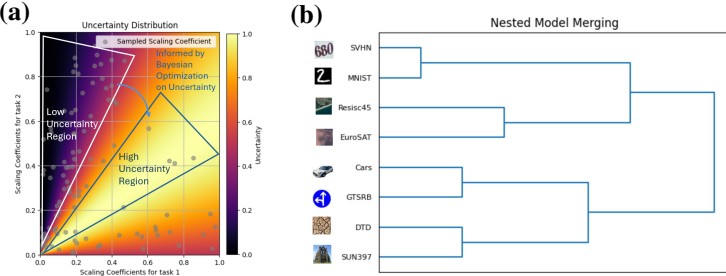

Figure 4: **(a)**: Utilize Bayesian optimization to guide the sampling of scaling coefficients according to the uncertainty distribution; **(b)**: An example of nested model merging for $N = 8$ models.

## 4 EXPERIMENTS

**Datasets and models**   We evaluate multi-task model merging on eight zero-shot image classification datasets following (Ilharco et al., 2022): SUN397 (Xiao et al., 2016), Cars (Krause et al., 2013), GT-SRB (Stallkamp et al., 2011), MNIST (LeCun, 1998), EuroSAT (Helber et al., 2019), SVHN (Netzer et al., 2011), DTD (Cimpoi et al., 2014), and RESISC45 (Cheng et al., 2017). We use the ViT-B/32 architecture in CLIP (Radford et al., 2021) as the pre-trained model for the experiments on vision tasks discussed in the main text. We show the results of these experiments in the main pages.

The rest of the experiments are presented in the Appendix due to page limits. The results from the datasets and tasks we experimented on are as follows: a zero-shot medical chest X-ray image classification task to show our model merging scheme works in a real-world application in the medical domain (Wang et al., 2017) (Appendix D.3); four fine-tuned Llama3 models in different languages: French, Arabic, Chinese, and Japanese (Appendix D.4)[2]; and three vision classification tasks using the ResNet18 (He et al., 2016) architecture: CIFAR-10 (Krizhevsky et al., 2009), Flowers-102 (Nilsback & Zisserman, 2008), and GTSRB (Stallkamp et al., 2011) to show that our model merging scheme also works with other model architectures (Appendix D.5).

---

[2]All the fine-tuned language Llama3 models can be found on Hugging Face. The IDs of the models are: French: jpacifico/French-Alpaca-Llama3-8B-Instruct-v1.0; Arabic: MohamedRashad/Arabic-Orpo-Llama-3-8B-Instruct; Chinese: shenzhi-wang/Llama3-8B-Chinese-Chat; Japanese: haqishen/Llama-3-8B-Japanese-Instruct.

## 4.1 BASELINE AND METRICS

**Pareto front-finding baselines** One of our baseline methods for obtaining Pareto fronts is the brute-force direct search, which can be regarded as the gold standard when the number of tasks is low ($N < 4$). This is because we can sample enough grid points of $\mathbf{c}$ and query the corresponding evaluation metrics $\{M_n(\theta(\mathbf{c}))\}_{n=1}^N$. We can then directly use the resulting evaluation metrics to find the Pareto front by direct comparisons of each task's performance of the merged model by $c_i \in \mathbf{c}$.

Another baseline method for finding Pareto fronts is the Multi-Objective Evolutionary Algorithm based on Decomposition (MOEA/D) (Zhang & Li, 2007). Key hyperparameters in MOEA/D include the population size and the number of generations, which significantly influence its performance by affecting the quality and diversity of the resulting Pareto front as well as the computational cost.

It is important to note that when the number of tasks grows, the required number of $(\mathbf{c}, \{M_n(\theta(\mathbf{c}))\}_{n=1}^N)$ pairs grows exponentially for the brute force method, as well as for the MOEA/D method, which is much larger than the points we can evaluate given the computational constraint. Thus, when the number of tasks is high ($N \geq 4$), the results from the brute-force direct search can no longer be considered ground truth. In Appendix E.2.2, we show an illustration of why the curse of dimensionality occurs when the number of tasks increases.

**Single merged model baselines** In addition to the brute-force method and MOEA/D, we compare with other model merging methods including SLERP (Shoemake, 1985), TIES-merging (Yadav et al., 2024), Task Arithmetic with a single scalar (Ilharco et al., 2022), Task Arithmetic with preferences as scalars (Ilharco et al., 2022), DARE combined with Task Arithmetic (Yu et al., 2023; Ilharco et al., 2022), and DARE combined with TIES-merging (Yu et al., 2023; Yadav et al., 2024).

**Win rate** We used the win rate to measure how often the Pareto front found by MAP outperforms the Pareto front found by the baseline in terms of multiple objectives. Let $PF_{MAP}$ and $PF_{baseline}$ represent the set of solutions in the Pareto fronts obtained from the MAP and the baseline methods, respectively. Each solution in these sets is a vector in $\mathbb{R}^N$, where $N$ is the number of objectives or tasks. We sampled $K = 100$ points from the decision space of each of the two Pareto fronts, denoted as $\mathbf{c}_k^{MAP}$ and $\mathbf{c}_k^{baseline}, k = 1, \ldots, K$. Then, we compared $M_n(\theta(\mathbf{c}_k^{MAP}))$ and $M_n(\theta(\mathbf{c}_k^{baseline}))$ pairwise for $k = 1, \ldots, K$ and $n = 1, \ldots, N$, resulting in $K^2 N$ comparisons. The ratio of instances where $M_n(\theta(\mathbf{c}_k^{MAP})) > M_n(\theta(\mathbf{c}_k^{baseline}))$ is computed as the win rate of $PF_{MAP}$:

$$\text{Win Rate} = \frac{1}{K^2 N} \sum_{i=1}^K \sum_{j=1}^K \sum_{n=1}^N \mathbb{I}\left[M_n(\theta(\mathbf{c}_i^{\text{MAP}})) > M_n(\theta(\mathbf{c}_j^{\text{baseline}}))\right]$$

where $K = 100$, $\mathbf{c}_i^{\text{MAP}} \in$ Decision Space of $PF_{\text{MAP}}, i = 1, \ldots, K$, $\mathbf{c}_j^{\text{baseline}} \in$ Decision Space of $PF_{\text{baseline}}, j = 1, \ldots, K$, $\mathbb{I}[\cdot]$ is the indicator function.

## 4.2 WIN RATE OF MAP OVER THE DIRECT SEARCH METHOD

Table 2 shows the results for the win rate of MAP over the brute force direct search method **when the number of tasks is small** ($N < 4$). In such cases, the Pareto front can be regarded as the ground truth Pareto front, and we can only query 30 and 50 scaling coefficients to achieve similar performance with the ground truth Pareto front.

Table 3 shows the results for the win rate of MAP over the brute force direct search method **when the number of tasks is high** ($N \geq 4$). The Pareto front generated by the brute force method can no longer be considered ground truth due to the limited number of points per dimension that is covered. In this setting, MAP performs much better than the brute-force (not ground truth) Pareto solutions.

Table 2: Win rate of the amortized PF over the brute-force direct search PF, where the latter can be regarded as the ground truth when $N < 4$. # $\mathbf{c}$ is the number of scaling coefficient vectors each algorithm evaluated to find the Pareto front. # $\mathbf{c}$ per dim = (# $\mathbf{c}$ direct search)$^{1/N}$ measures the sparsity of points the direct search method uses, as illustrated in Figure 11.

| N | # $\mathbf{c}$ (direct search) | # $\mathbf{c}$ per dim | # $\mathbf{c}$ (MAP) | Win rate (MAP) | $R^2$ (MAP) |
|---|---|---|---|---|---|
| 2 | 200 | 14.14 | 30 | 49.81% (±0.30) | 0.953 (±0.018) |
| 3 | 300 | 6.69 | 50 | 46.90% (±0.71) | 0.980 (±0.003) |

Table 3: Win rate of the amortized PF over the brute-force direct search PF, where the latter can **no longer** be regarded as the ground truth when $N \geq 4$. # of **c** is the number of scaling coefficient vectors each algorithm evaluated to find the Pareto front. # **c** per dim = (# **c** direct search)$^{1/N}$ measures the sparsity of points the direct search method uses, as illustrated in Figure 11.

| N | # **c** (direct search) | # **c** per dim | # **c** (MAP) | Win rate (MAP) | $R^2$ (MAP) |
|---|---|---|---|---|---|
| 4 | 300 | 4.16 | 60 | 50.67% ($\pm$2.44) | 0.984 ($\pm$0.004) |
| 5 | 500 | 3.47 | 85 | 53.00% ($\pm$1.88) | 0.941 ($\pm$0.019) |
| 6 | 500 | 2.82 | 100 | 60.71% ($\pm$1.34) | 0.941 ($\pm$0.030) |
| 7 | 1000 | 2.68 | 140 | 63.42% ($\pm$1.91) | 0.891 ($\pm$0.024) |
| 8 | 1000 | 2.37 | 250 | 65.58% ($\pm$0.94) | 0.868 ($\pm$0.028) |

### 4.3 COMPARING WITH OTHER SINGLE MERGED MODEL BASELINE METHODS

We compared the performance of MAP with TIES-merging (Yadav et al., 2023), TIES-merging with DARE (Yu et al., 2023), Task Arithmetic with DARE, Task Arithmetic with normalized preference as scaling coefficients (Ilharco et al., 2023), Ada-merging++ (Yang et al., 2024), DELLA-merging (Deep et al., 2024) and SLERP (Shoemake, 1985). For dimension 2, we can directly visualize the results, as shown in Figure 5. The solutions found by MOEA/D exhibited clustering and a lack of adequate spread across the Pareto front. In addition, the computational cost of MOEA/D was significantly higher than that of MAP. For example, with a population size of 50 and 20 generations (which produced the best results for MOEA/D in terms of diversity as shown in Figure 5), the total number of evaluations amounted to 2500 for two tasks. In comparison, MAP achieved its results with only 60 total evaluations—a reduction of over 95%. This highlights the computational efficiency of MAP in obtaining a diverse and high-quality Pareto front.

**MAP offers a well-distributed set of Pareto optimal solutions**   None of the baseline methods can directly dominate all Pareto solutions found by MAP, and most of them lie either within or on the Pareto front found by MAP. In addition, we sampled 10 points from the predicted Pareto front generated by MAP and evaluated them (MAP Pareto solutions, real) to confirm that they indeed lie close to the predicted Pareto front. This evidence further confirms the usefulness of MAP as a general strategy for finding a diverse and well-distributed set of Pareto solutions that none of the existing model merging methods can substitute.

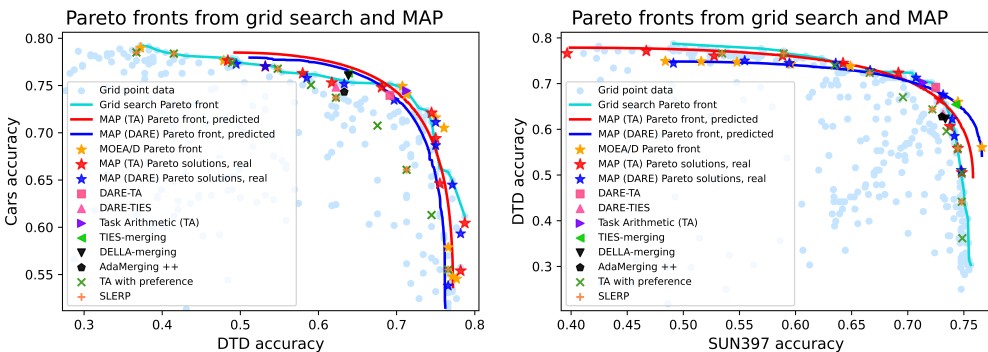

Figure 5: The Pareto fronts obtained using MAP with Task Arithmetic, MAP with Task Arithmetic and DARE. We sampled 10 Pareto solutions from the predicted front by MAP and evaluated them to obtain the real values. We plotted the results obtained using TIES-merging, Task Arithmetic (TA) with a single scalar for all tasks, Task Arithmetic with preferences as scalars, TA combined with DARE (DARE-TA), TIES-merging combined with DARE (DARE-TIES), and SLERP.

**MAP is an out-of-the-box plugin for other task-vector based model merging methods**   As shown in Figure 5, MAP is a plug-in that can be directly combined with other model merging methods, allowing users to control their preferences over each task using scaling coefficients. For example, MAP can be combined with DARE, Task Arithmetic, etc.

For dimensions higher than 2, we use preference-weighted evaluation sum to compare the Pareto front found by MAP with other baseline methods. We sample 20 preference vectors and normalize

them. Then, we pick the solution by MAP that maximizes the preference-weighted sum of accuracies and compare it with those of the baseline methods. The results are shown in Table 4. We observe that even in higher dimensions, when users have specific preferences, MAP remains effective and can better accommodate those preferences. Note that when using MAP, users are not required to pre-specify their preference vector. In fact, users can obtain the entire Pareto front to better understand the trade-offs. This preference vector-based metric is solely for evaluation.

Table 4: We compared MAP with a set of baseline methods by sampling a set of 20 normalized preference vectors and computing the preference-weighted sum of accuracies. ↑ indicates higher is better. The number after ± is the standard deviation. Please refer to Table 2 and Table 3 for the number of scaling coefficient vectors used for different numbers of tasks.

| | Preference weighted sum of accuracies (↑) | | | | | | |
|---|---|---|---|---|---|---|---|
| # tasks | 2 | 3 | 4 | 5 | 6 | 7 | 8 |
| Single task models | 75.84±1.76 | 77.03±1.84 | 82.43±4.40 | 87.69±4.50 | 88.52±4.02 | 89.26±3.58 | 90.62±2.52 |
| MTL | 73.63±0.30 | 75.13±1.00 | 80.10±2.79 | 84.93±3.58 | 86.78±2.94 | 87.40±2.56 | 89.11±2.36 |
| Model soups ((Wortsman et al., 2022)) | 67.79±1.46 | 64.25±2.15 | 66.04±3.22 | 67.01±3.42 | 63.11±1.99 | 63.35±2.17 | 64.36±2.77 |
| TIES-merging ((Yadav et al., 2024)) | 69.30±0.33 | 67.60±0.58 | 71.79±2.93 | 76.49±3.10 | 73.74±2.96 | 72.54±2.87 | 72.24±1.91 |
| DARE-TIES | 67.62±1.65 | 66.49±2.34 | 71.39±4.45 | 74.55±4.55 | 73.34±4.10 | 71.43±3.84 | 71.89±2.86 |
| Task Arithmetic ((Ilharco et al., 2022)) | **70.73±1.84** | 61.15±2.33 | 52.69±4.23 | 61.58±4.62 | 51.37±3.84 | 39.79±3.97 | 60.77±2.84 |
| TA with preference as weights | 69.22±1.4 | 66.88±2.37 | 68.73±5.48 | 71.92±5.5 | 68.13±4.69 | 68.14±4.2 | 68.17±2.89 |
| DARE-TA | 70.61±0.22 | 64.18±1.24 | 58.04±8.19 | 65.39±7.03 | 56.76±7.01 | 46.75±5.73 | 64.51±3.81 |
| Ada-Merging++ ((Yang et al., 2024)) | 67.27±1.92 | 67.13±1.92 | 71.19±4.43 | 76.84±4.71 | 74.13±4.07 | 72.58±4.16 | 72.55±2.83 |
| DELLA-Merging ((Deep et al., 2024)) | 67.10±2.08 | 65.92±2.48 | 70.71±4.31 | 74.43±4.32 | 72.64±3.77 | 71.16±3.95 | 71.49±2.83 |
| MOEA/D | 70.22 ± 1.46 | 67.94 ± 1.79 | 70.85 ± 4.58 | 72.03 ± 4.03 | 67.88 ± 3.09 | 69.06 ± 2.97 | 68.59 ± 2.89 |
| MAP | 70.70±1.76 | **69.05±1.84** | **72.84±4.40** | **77.31±4.50** | **74.26±4.02** | **73.40±3.58** | **72.96±2.52** |

## 5 RELATED WORK

Due to space constraints, for a more comprehensive discussion, please refer to Appendix B.

**Multi-objective optimization** Multi-objective optimization identifies diverse Pareto solutions with different trade-offs. The multi-task problem has been approached from a MOOP perspective (Sener & Koltun, 2018; Lin et al., 2019), utilizing algorithms like MGDA, IMTL, GradNorm, RLW, and scalarization. While these methods iteratively optimize $\mathbb{R}^d$ model parameters with significant computational cost, our approach performs post-training MOOP over $\mathbb{R}^N$ scaling coefficients, reducing computational burden. For $N \geq 3$ objectives, MOOP becomes many-objective optimization, challenging traditional algorithms (e.g. NSGA-II and SPEA2). Advanced methods such as $\epsilon$-MOEA (Deb et al., 2003), MSOPS (Hughes, 2005), SMS-EMOA (Beume et al., 2007), and NSGA-III (Deb & Jain, 2013), or the KKT approach (Augusto et al., 2014) are required.

**Task arithmetic** Task arithmetic, a model merging method gaining attention, uses weighted averages of models for multi-task performance. Various approaches exist for selecting scaling coefficients: Ilharco et al. (2022) used equal scaling, which is suboptimal and limited; Yang et al. (2023) optimized weights using Shannon entropy but required unlabeled test data, which conflict with data privacy goals in model merging. Beyond task arithmetic, Ainsworth et al. (2023) merge models via weight matrix permutations, even without shared pretraining. Jin et al. (2023) introduced dataless knowledge fusion. Daheim et al. (2024) proposed uncertainty-based gradient matching for improved merging performance and robustness.

## 6 CONCLUSION AND LIMITATIONS

We introduced MAP, a novel low-compute approach to efficiently merge models while accounting for trade-offs between multiple tasks. By leveraging a quadratic approximation of the evaluation metrics, MAP successfully identifies amortized Pareto fronts without the need for gradient descent for deep neural networks. The algorithm's efficiency is further enhanced through Bayesian adaptive sampling and nested merging, enabling it to scale to problems with a higher number of tasks. Moreover, MAP is an out-of-the-box plug-in for other task-vector-based model merging methods.

However, we would like to point out some limitations of our method. We do not have a detector algorithm to detect if two tasks have proper trade-offs. Approximating a Pareto "front" with only a single Pareto solution can lead to degenerate solutions. In this case, practitioners should be informed.

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

# Contents

## A  NOTATIONS

For clarity, we summarize the notations used throughout the paper in Table 5.

Table 5: Table of notations.

| Notations | Explanation |
|---|---|
| $N$ | Number of tasks |
| $\mathbf{c}$ | Vector of scaling coefficients for task vectors |
| $c_n$ | Scaling coefficients for the $n$th task |
| $\boldsymbol{\theta}_{\text{pre}}$ | Pretrained model parameters |
| $\theta_m(\mathbf{c})$ | Merged model parameters as a function of scaling vector $\mathbf{c}$ |
| $\mathbf{v_n}$ | Task vector for task $n$ |
| $\mathbf{V}$ | Task matrix formed by concatenating task vectors |
| $M_n(\mathbf{c})$ | Evaluation metric for task $n$ as a function of $\mathbf{c}$ |
| $\mathbf{A}_n$ | Quadratic coefficient matrix in the surrogate model |
| $\mathbf{b}_n$ | Linear coefficient vector in the surrogate model |
| $e_n$ | Constant term in the surrogate model |
| $\tilde{M}_n(\mathbf{c}; \mathbf{A}_n, \mathbf{b}_n, e_n)$ | Surrogate quadratic model for task $n$ |
| $\Omega$ | Set of scaling coefficient vectors $\mathbf{c}$ used for estimating surrogate model parameters |
| $K$ | Number of sampled coefficient vectors in set $\Omega$ |

## B  MORE DISCUSSION ON RELATED WORK

**Second-order Taylor expansion**  In deep learning, the second-order Taylor expansion and thus the quadratic approximation on the evaluation metric is an important technique, that characterizes the metric landscape. For example, the Newton-Ralphson method is derived from it: $\mathbf{w}_t - \mathbf{w}_{t+1} = \mathbf{H}^{-1}\mathbf{G} = \text{argmin}_{\mathbf{v}} M(\mathbf{w}_t - \mathbf{v})$ given that $M(\mathbf{w}_t - \mathbf{v}) = M(\mathbf{w}_t) - \mathbf{v}\mathbf{G} + \frac{1}{2}\mathbf{v}^{\top}\mathbf{H}\mathbf{v}$. The quadratic approximation is also combined with Lipschitz smoothness ($L$) in the classic convergence analysis of SGD (Bubeck et al., 2015; Ghadimi & Lan, 2013), through $M(\mathbf{w}_t - \eta\mathbf{G}) \leq M(\mathbf{w}_t) - \eta||\mathbf{G}||^2 + \frac{L\eta^2}{2}||\mathbf{G}||^2$. Interestingly, although the approximation is only accurate in the local neighborhood, it can be used to indicate the global convergence over the iterations. One reason is that state-of-the-art neural networks are usually over-parameterized and undergo lazy training, meaning that the converging parameters are close to the initialization (Chizat et al., 2019; Du et al., 2019; Allen-Zhu et al., 2019). Thus, the local approximation informs the global convergence behavior. In particular, this approximation has played important roles in the scaling laws of neural networks (e.g. Equation 3.3 in (Su et al., 2024)) that predict the performance and help select hyperparameters before training actually takes place.

**Pareto fronts for Multi-task Learning**  Recent advancements in multi-task learning (MTL) have seen diverse approaches to Pareto Front learning in machine learning. Sener & Koltun (2018) explicitly formulated multi-task learning as a multi-objective optimization problem and adapted gradient-based multi-objective optimization algorithms to large-scale learning problems. Lin et al. (2019) developed a constrained optimization approach to find multiple Pareto optimal solutions representing different trade-offs among tasks in multi-task learning and solved decomposed sub-problems in parallel using gradient-based optimization. It requires separate training runs for each solution. Navon et al. (2020) and Lin et al. (2020) employed hypernetworks for continuous Pareto Front approximation, generating target network weights based on preferred trade-offs. However, the size of these hypernetworks can become prohibitively large and needs to be properly trained as well. Ruchte and Grabocka Ruchte & Grabocka (2021) introduced a memory-efficient method by augmenting network inputs with desired trade-offs, although this may obscure the network's conditioning due to nonlinear dynamics, and it is also based on the gradient updating method.

**Model-merging Applications in LLM**  In the realm of model merging applications for language model preferences, recent research has made significant progress. Ramé et al. (2023) and Jang et al. (2023) introduced "rewarded soups" and "personalized soups", which utilize model soup to interpolate weights of networks fine-tuned on diverse rewards to achieve Pareto-optimal generalization across

preference spaces and address the challenge of aligning large language models with individual perspectives. WARP (Weight Averaged Rewarded Policies) (Ramé et al., 2024) and WARM (Weight Averaged Reward Models) (Ram'e et al., 2024) demonstrate how merging policies or reward models in weight space can refine the trade-off between competing constraints, such as reward optimization and alignment with pre-trained knowledge or human preferences. (Zhong et al., 2024) developed "Panacea", which reframes alignment as a multi-dimensional preference optimization problem, using singular value decomposition-based low-rank adaptation to guide model behavior with a low-dimensional preference vector. To address the limitations of scalar rewards in RLHF, Zhou et al. (2023) introduced Multi-Objective Direct Preference Optimization, an RL-free algorithm that extends Direct Preference Optimization to handle multiple alignment objectives efficiently. Du et al. (2024) relies on "mutation" and crossover operations typical of evolutionary algorithms, exploring the parameter space by combining and perturbing existing solutions. Models are evaluated on development datasets, and only those that improve upon their predecessors are retained, guiding the population toward better-performing solutions. Li et al. (2024) employs Bayesian optimization with a weak-to-strong approach and utilizes Fisher information to improve the selection of configurations for evaluation, aiming to find optimal merging configurations within limited computational budgets. Wang et al. (2024b) investigated online learning with limited resources, and successfully deployed model aggregation to handle changing environments, establishing optimal theoretical guarantees. Finally, Wang et al. (2024a) proposed the Directional Preference Alignment framework, which incorporates multi-objective reward modeling to represent user preferences, offering intuitive arithmetic control over LLM generation for diverse user preferences.

**Bayesian Optimization**   Bayesian optimization has been widely used in scenarios that require efficient exploration of parameter spaces, particularly when evaluating the performance of each configuration is costly or time-consuming. This method is especially advantageous in machine learning and hyperparameter tuning, where traditional optimization techniques may be computationally prohibitive (Wilson et al., 2017; Jain et al., 2022; Moss et al., 2020; Pyzer-Knapp, 2018; Terayama et al., 2021).

## C   ADDITIONAL DETAILS ON THE METHODS

### C.1   PROOF: NEGLIGIBILITY OF THE REMAINDER IN MULTIVARIATE TAYLOR SERIES

**Corollary 2.** *If $f : \mathbb{R}^n \to \mathbb{R}$ is $(k + 1)$ times continuously differentiable in a neighborhood around a point $a \in \mathbb{R}^n$, then the Taylor polynomial $T_k(x)$ of order $k$ provides an accurate approximation of $f(x)$ when $x$ is sufficiently close to $a$. Furthermore, the remainder term $R_k(x)$ becomes negligibly small as $\|x - a\|$ approaches zero, assuming that the $(k + 1)$th derivatives of $f$ are bounded by a constant $M$ in the neighborhood between $a$ and $x$.*

**Proof**   *Consider the Taylor series expansion of $f$ around the point $a$, truncated at order $k$:*

$$T_k(x) = \sum_{|\alpha| \leq k} \frac{D^\alpha f(a)}{\alpha!} (x - a)^\alpha$$

*where $\alpha$ is a multi-index of non-negative integers, $D^\alpha f(a)$ denotes the partial derivatives of $f$ at $a$ corresponding to $\alpha$, and $(x - a)^\alpha = (x_1 - a_1)^{\alpha_1} \ldots (x_n - a_n)^{\alpha_n}$.*

**Assumptions**

1. *Proximity: $\|x - a\| \to 0$ where $\| \cdot \|$ denotes the Euclidean norm in $\mathbb{R}^n$.*

2. *Bounded Derivatives: There exists a constant $M$ such that for all multi-indices $\alpha$ with $|\alpha| \equiv k + 1$, the norm of the tensor $D^\alpha f$ evaluated at any point $\xi$ between $a$ and $x$ is bounded by $M$.*

$$\|D^\alpha f(\xi)\| = \sup_{\|v_1\|=1, \ldots, \|v_{k+1}\|=1} |D^\alpha f(\xi)(v_1, \ldots, v_{k+1})| \leq M$$

*The remainder term of the Taylor series expansion is given by:*

$$R_k(x) = \sum_{|\alpha|=k+1} \frac{D^\alpha f(\xi)}{\alpha!} (x-a)^\alpha$$

*Given the assumptions, we estimate:*

$$|R_k(x)| \leq \sum_{|\alpha|=k+1} \frac{\|D^\alpha f(\xi)\|}{\alpha!} \|x-a\|^{k+1} \leq \sum_{|\alpha|=k+1} \frac{M}{\alpha!} \|x-a\|^{k+1}$$

*As $\|x-a\| \to 0$, the term $\|x-a\|^{k+1}$ goes to zero. Thus, the remainder term $R_k(x)$ becomes arbitrarily small, making it negligible.*

*In conclusion, under the stated assumptions, the Taylor series truncated at order $k$, $T_k(x)$, provides an accurate approximation of $f(x)$ near $a$, and the remainder $R_k(x)$ can be ignored as $\|x-a\| \to 0$ and the higher-order derivatives remain bounded by $M$.*

## C.2 CLOSED-FORM SOLUTION FOR SURROGATE MODEL PARAMETERS

**Corollary 3** (Closed-form Solution for Surrogate Model Parameters). *Under Assumption 1, for each task $n = 1, \ldots, N$, the optimization problem*

$$(A_n^*, \mathbf{b}_n^*, e_n^*) = \arg \min_{A_n, \mathbf{b}_n, e_n} \sum_{\mathbf{c} \in \Omega} \left| M_n\left(\theta_m(\mathbf{c})\right) - \tilde{M}_n(\mathbf{c}; A_n, \mathbf{b}_n, e_n) \right|^2$$

*is equivalent to solving the following linear regression problem:*

1. **Predictors**: *For each coefficient vector $\mathbf{c} = (c_1, c_2, \ldots, c_N)^\top \in \mathbb{R}^N$, construct the predictor vector $\mathbf{C}_n(\mathbf{c}) \in \mathbb{R}^{\frac{N(N+3)}{2}+1}$ as*

$$\mathbf{C}_n(\mathbf{c}) = [c_1^2, c_2^2, \ldots, c_N^2, c_1 c_2, c_1 c_3, \ldots, c_{N-1} c_N, c_1, c_2, \ldots, c_N, 1]$$

   *This vector includes:*

   - *Quadratic terms: $c_i^2$ for $i = 1, \ldots, N$.*
   - *Interaction terms: $c_i c_j$ for $1 \leq i < j \leq N$.*
   - *Linear terms: $c_i$ for $i = 1, \ldots, N$.*
   - *Constant term: 1.*

2. **Response Variable**: *Let $\mathbf{y}_n \in \mathbb{R}^K$ be the vector of observed evaluation metrics for task $n$ across all sampled coefficients $\Omega = \{\mathbf{c}^{(1)}, \ldots, \mathbf{c}^{(K)}\}$:*

$$\mathbf{y}_n = \begin{bmatrix} M_n\left(\theta_m(\mathbf{c}^{(1)})\right) \\ M_n\left(\theta_m(\mathbf{c}^{(2)})\right) \\ \vdots \\ M_n\left(\theta_m(\mathbf{c}^{(K)})\right) \end{bmatrix}$$

3. **Design Matrix**: *Construct the design matrix $\mathbf{C}_n \in \mathbb{R}^{K \times \left(\frac{N(N+3)}{2}+1\right)}$ where each row corresponds to $\mathbf{C}_n(\mathbf{c}^{(k)})^\top$ for $k = 1, \ldots, K$.*

4. **Coefficient Vector**: *Define the coefficient vector $\boldsymbol{\beta}_n \in \mathbb{R}^{\frac{N(N+3)}{2}+1}$ as*

$$\boldsymbol{\beta}_n = \begin{bmatrix} vec(A_n) \\ \mathbf{b}_n \\ e_n \end{bmatrix}$$

   *where $vec(A_n)$ represents the vectorization of the upper triangular part of $A_n$, including the diagonal elements.*

**Optimal Solution**: *The parameters $(A_n^*, \mathbf{b}_n^*, e_n^*)$ that minimize the mean squared error are obtained via the Ordinary Least Squares (OLS) solution:*

$$\boldsymbol{\beta}_n^* = \left(\mathbf{C}_n^\top \mathbf{C}_n\right)^{-1} \mathbf{C}_n^\top \mathbf{y}_n$$

**Interpretation**: *This closed-form solution provides the optimal surrogate model parameters by fitting a quadratic model to the observed evaluation metrics through linear regression. The design matrix $\mathbf{C}_n$ incorporates all necessary quadratic, interaction, linear, and constant terms, enabling the surrogate model $\tilde{M}_n(\mathbf{c}; A_n, \mathbf{b}_n, e_n)$ to accurately approximate $M_n(\theta_m(\mathbf{c}))$ within the specified region.*

# D  ADDITIONAL DETAILS ON EXPERIMENTS

## D.1  EXPERIMENT SETUP

Table 6: Experiment setup in terms of task details and models.

| Task type | Metric | # of total tasks | Model type |
|---|---|---|---|
| Zero-shot Classification (normal) | Accuracy | 8 | ViT-B/32 (CLIP) ((Dosovitskiy et al., 2020)) |
| Zero-shot Classification (medical) | Accuracy | 2 | ViT-B/32 (CLIP) ((Dosovitskiy et al., 2020)) |
| Language Generation | Loss/Perplexity | 4 | Llama3-8B ((Touvron et al., 2023)) |
| Image Classification | Accuracy | 3 | ResNet-18 ((He et al., 2016)) |

## D.2  ADDITIONAL METRIC: GENERATIONAL DISTANCE AND INVERTED GENERATIONAL DISTANCE

We evaluated the quality of the Pareto front in capturing the shape of the ground truth Pareto front by measuring how much the predicted Pareto front converges to the ground truth Pareto front by calculating the generational distance (GD) (Van Veldhuizen, 1999) and how much the predicted Pareto front covers the ground truth Pareto front by calculating the inverted generational distance (IGD) (Coello & Cortés, 2005). GD and IGD are standard measures used in evolutionary multi-objective optimization to evaluate the solutions found by the evolutionary algorithms. Given a predicted Pareto solution sets $PF_1 = \{(\tilde{M}_1^j(\theta_{merged}(\mathbf{c})), \ldots, \tilde{M}_N^j(\theta_{merged}(\mathbf{c}))) \mid j = 1, 2, \ldots, K_1\}$, and a ground truth Pareto solution sets $PF_2 = \{(M_1^j(\theta_{merged}(\mathbf{c})), \ldots, M_N^j(\theta_{merged}(\mathbf{c}))) \mid j = 1, 2, \ldots, K_2\}$ the GD and IGD metrics are defined as

$$GD(PF_1) \equiv \frac{1}{K_1} \left(\sum_{i=1}^{K_1} d_i^p\right)^{1/p} \text{ and } IGD(PF_1) \equiv \frac{1}{K_2} \left(\sum_{i=1}^{K_2} \tilde{d}_i^p\right)^{1/p}$$

where $d_i$ is the minimal Euclidean distance from a specific Pareto Solution in $PF_1$ to $PF_2$ and $\tilde{d}_i$ is the minimal Euclidean distance from a specific Pareto Solution in $PF_2$ to $PF_1$.

Since the Pareto front that resulted from the direct search method when the number of tasks is low (i.e. strictly smaller than 4) can be deemed as ground truth Pareto fronts, we computed GD and IGD to compare the Pareto fronts produced by MAP against those obtained via brute-force grid search.

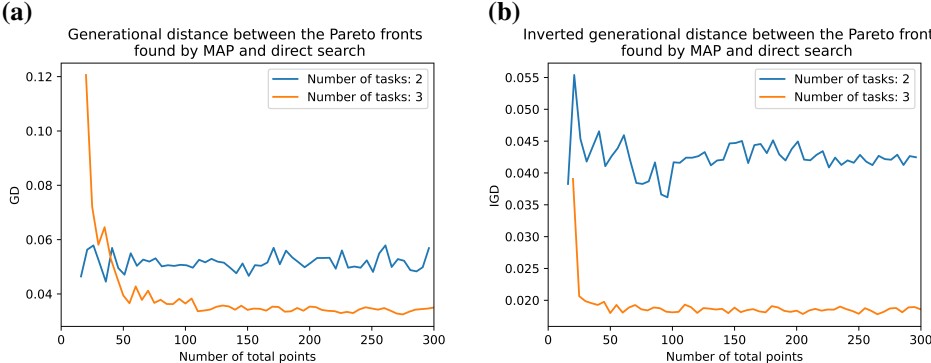

Figure 6: (a) Generational distance between the Pareto fronts by MAP and by direct search for dimensions 2 and 3. (b) Inverted generational distance between the Pareto fronts by MAP and by direct search for dimensions 2 and 3. For both subfigures, the x-axis is the number of total scaling coefficients used by MAP. For dimension 2, direct search used 200 scaling coefficients, and 300 for dimension 3.

### D.3 ZERO-SHOT MEDICAL IMAGE CLASSIFICATION

In addition to natural images, we used another dataset consisting of over 112,000 chest X-rays and 30,000 unique patients (of Health et al., 2017). It originally contained 15 classes (14 diseases and 1 class for no finding). We split the dataset into two groups, where medical task 1 specifically tries to classify Atelectasis, Consolidation, Infiltration, Pneumothorax, and medical task 2 tries to classify Nodule, Mass, and Hernia. An example image taken from the dataset is shown in Figure 7 (a).

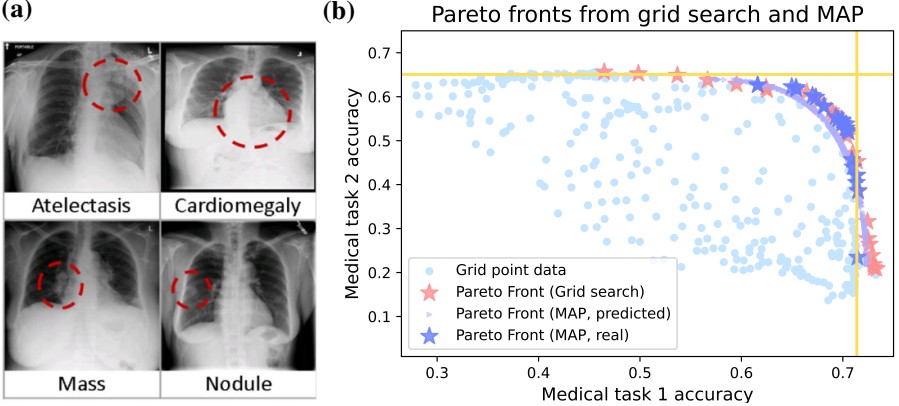

Figure 7: (a) Example figure from the NIH (of Health et al., 2017) dataset. (b) Pareto fronts found by brute-force direct search using 400 points and by MAP using 30 points. We randomly sampled 25 points from the predicted Pareto front by MAP. The resulting IGD is 0.016, and GD is 0.014.

### D.4 MERGING LANGUAGE MODEL

**Merging Languages** We merge Llama-3-8B language models in French[3] and Arabic[4], as well as Chinese[5] and Japanese[6], as shown in Table 7. The ground truth Pareto front, however, exhibits significant limitations. It contains only a sparse set of points, indicating that few merged models dominate the rest. This also suggests that the trade-off between metrics across different languages

---

[3]https://huggingface.co/jpacifico/French-Alpaca-Llama3-8B-Instruct-v1.0
[4]https://huggingface.co/MohamedRashad/Arabic-Orpo-Llama-3-8B-Instruct
[5]https://huggingface.co/shenzhi-wang/Llama3-8B-Chinese-Chat
[6]https://huggingface.co/haqishen/Llama-3-8B-Japanese-Instruct

is not substantial. Despite these challenges, our algorithm successfully identifies the Pareto fronts. Please refer to Table 7 and Appendix D.4 for GD and IGD metrics between the Pareto fronts find by Grid Search and find by MAP and a visualization of the Pareto fronts found by MAP.

Furthermore, we experimented with merging Arabic+French and Chinese+Japanese using the nested scheme described in Algorithm 2, while considering various preference settings. However, the resulting Pareto front typically contains only a single model. This sparsity hinders our ability to effectively characterize a meaningful Pareto front in such scenarios.

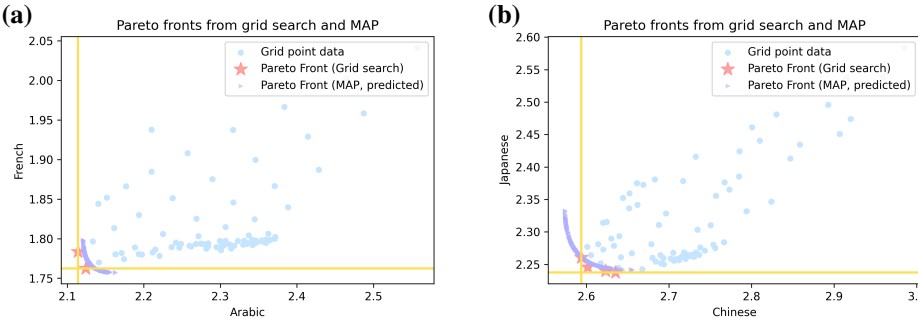

Figure 8: (a) Amortized Pareto front by MAP on merging Llama-3 fine-tuned on French with Llama-3 fine-tuned on Arabic; (b) Amortized Pareto front on merging Llama-3 fine-tuned on Chinese with Llama-3 fine-tuned on Japanese. The yellow line represents the evaluation metric of the corresponding single task model.

**Merging Language Models Specializing in Math and Code**   Following DARE ( (Yu et al., 2023)), we also experimented with merging two language models each with 13B parameters specializing in mathematics (referred to as the math model) and coding (referred to as the code model). The base model used is Llama-2 13B[7], the math model is WizardMath-13B[8], and the code model is llama-2-13b-code-alpaca[9]. Both the math and code models are fine-tuned versions of Llama-2 13B.

The primary reason for selecting Llama-2 as the base model, rather than newer models like Llama-3 or Qwen-2.5, is the lack of suitable math and code models fine-tuned from the same base configuration. For instance, although Qwen-2.5 provides both Qwen-2.5-Math and Qwen-2.5-Coder, the two models differ significantly in their configurations, specifically, the rope theta parameter is set to 10,000.0 for the math model and 1,000,000.0 for the coder model. This discrepancy makes it challenging to effectively merge these models. Please refer to Figure 9 for a visualization of the Pareto front obtained from MAP and Table 7 for GD and IGD metrics between the Pareto fronts obtained from Grid Search and from MAP.

Table 7: The GD and IGD metrics score for Arabic+French, Chinese+Japanese and Math+Code between the Pareto fronts find by Grid Search and find by MAP.

|  | GD | IGD | GD+IGD |
|---|---|---|---|
| Arabic+French | $0.023_{0.010}$ | $0.035_{0.018}$ | $0.058_{0.028}$ |
| Chinese+Japanese | $0.014_{0.013}$ | $0.028_{0.017}$ | $0.041_{0.026}$ |
| Math+Code | $0.039_{0.009}$ | $0.0.018_{0.002}$ | $0.057_{0.008}$ |

---

[7] https://huggingface.co/meta-llama/Llama-2-13b-hf
[8] https://huggingface.co/vanillaOVO/WizardMath-13B-V1.0
[9] https://huggingface.co/layoric/llama-2-13b-code-alpaca

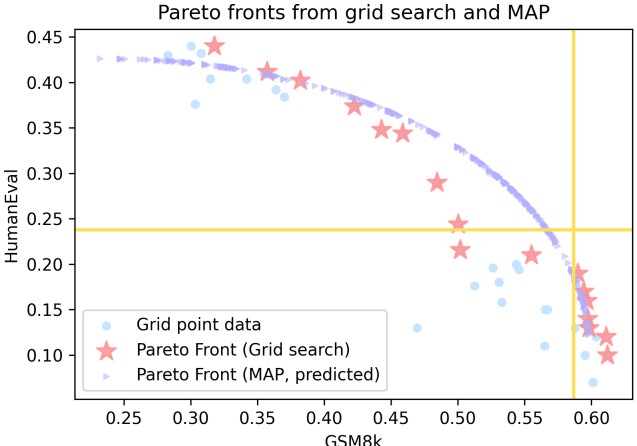

Figure 9: Amortized Pareto front by MAP on merging WizardMath-13B and code-alpaca-13B. The yellow line represents the evaluation metric of the corresponding single task model.

## D.5   ADDITIONAL EXPERIMENT RESULTS ON RESNET

We performed additional experiments on ResNet18 (He et al., 2016) by merging two models finetuned on CIFAR10 (Krizhevsky et al., 2009) and Flowers102 (Nilsback & Zisserman, 2008) and show the obtained Pareto front in Figure 10. Unlike ViT models, which perform zero-shot classification, ResNet requires additional fine-tuning of the classification head after model merging. We demonstrate that our method still applies to those models.

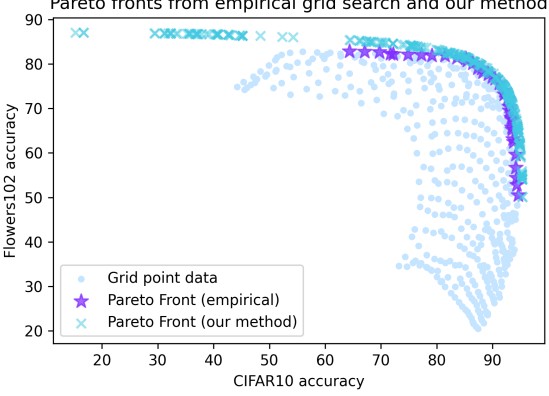

Figure 10: Pareto front obtained for two ResNet18 models on CIFAR-10 and Flowers-102. We perform additional finetuning of the classification head after merging the model weights.

# E    MORE DETAILS ON ALGORITHMS

## E.1    PLAIN MAP

It is important to note that Plain MAP (Algorithm 1) is an out-of-the-box plugin-like method. Many parts of it are kept generic: there are multiple ways to sample $\Omega$ in terms of the number of $\mathbf{c}$ and the style of sampling (Bayesian or not), thus creating a tradeoff between the quality of the Pareto front and the computational time; the task vector can be computed on the more memory-capable CPU and possibly in a parameter-efficient manner (e.g. LoRA, Adapter, and BiTFiT): for example, computing $\mathbf{v}_n$ via the subtraction of two full 7B models requires $2 \times 23 = 46$GB of memory capacity, but the memory cost for PEFT models may reduce to 16MB by only subtracting the non-frozen components; the for-loops in lines 2 and 4 can be computed in parallel, possibly using a much smaller subset of the data, and thus enjoying a computational speed-up.

## E.2    NESTED-MERGING MAP

### E.2.1    OVERVIEW

In this section, we explain the procedure of the algorithm in Figure 4 in detail.

Empirically, we only need 20 scaling coefficients to get a Pareto front of good quality to merge 2 tasks. However, due to the curse of dimensionality, we need exponentially more pairs of $(\mathbf{c}, \{\tilde{M}_n(\theta_m(\mathbf{c}))\}_{n=1}^N)$ pairs to achieve a good quality of the Pareto front in the case of 8 tasks. Theoretically, in the best case, the number of points required to obtain a good quality Pareto front for $N$ tasks is $O(N^3)$. In the worst case, when the curse of dimensionality dominates, it could be $O(N \cdot 2^N)$. Using the nested merging scheme, we reduce the computation complexity from $O(N \cdot 2^N)$ to $O(N \log N)$. Please refer to Table 8 for the detailed computational complexity comparison. Algorithm details are presented in Algorithm 2.

---

**Algorithm 2** Nested-merging MAP

---

**Input:**    A predetermined preference $pref \in \mathbb{R}^N$ over the $N$ tasks, the tuple of task, loss, task vector:
$\qquad G_n = (\text{task}_n, l_n, \boldsymbol{\theta}_{ft}^n)$
Normalize $pref$ to make sure the sum is 1
Initialize the set $\tau = \{G_1, \ldots, G_N\}$
**while** $|\tau| > 1$ **do**
$\qquad$ Find the pair of $(G_i, G_j) \in \tau$ that are closest to each other in terms of $(l_i, l_j)$
$\qquad$ Implement Algorithm 1 to find the Pareto front $\text{PF}_{i,j}$ between $(\tilde{M}_i, \tilde{M}_j)$
$\qquad$ Select $\mathbf{c}^* = (c_i^*, c_j^*) \in \mathbb{R}^2$ based on the Pareto front $\text{PF}_{i,j}$
$\qquad$ Merge the models by $\boldsymbol{\theta}_{\text{merge}}^{i,j} = \boldsymbol{\theta}_{\text{pre}} + c_i(\boldsymbol{\theta}_{\text{pre}} - \boldsymbol{\theta}_{ft}^i) + c_j(\boldsymbol{\theta}_{\text{pre}} - \boldsymbol{\theta}_{ft}^j)$
$\qquad$ Calculate the weighted average loss on the two tasks $l_{ij} = pref_i l_i + pref_j l_j$
$\qquad$ Update $\tau$ by replacing $\{G_i, G_j\}$ with $\{G_{ij}\}$, where $G_{ij} \equiv (\text{task}_{i,j}, l_{ij}, \boldsymbol{\theta}_{\text{merge}}^{i,j})$
**end**
**return** $\boldsymbol{\theta}_{merge}^{1,2,\ldots,N}$

---

### E.2.2    TIME COMPLEXITY OF NESTED-MERGING MAP

Table 8: Computational cost of MAP variants for $N$ models.

|  | # evals per task | minimum # evals (total) | # evals (total) |
|---|---|---|---|
| Plain MAP | $O(N^2)$ | $O(N^3)$ | $O(N \cdot 2^N)$ |
| Nested-merging MAP | $O(1)$ | $O(N \log N)$ | $O(N \log N)$ |

To estimate the computational complexity of nested-merging MAP, we denote N as the number of tasks. The number of total evaluations needed for nested-merging MAP is: $N/2 \times 2 + \ldots + N/2^m \times 2^m = O(N \log N)$ where $2^{m-1} < N \le 2^m$.

Detailed calculations are as follows. When the number of tasks is 8, in the first rounds, four 2-task merging procedures are running in parallel. Each of the procedures evaluates 2 tasks. In the procedure of model-merging between 2 tasks, as discussed, we need to sample 20 scaling coefficients $\mathbf{c}$ and evaluate 20 merged models on the 2 tasks. Thus, it takes $4 \times 20 \times 2 = 160$ times of evaluation in the first round. In the second round, 2 2-task merging procedures are running in parallel, each of them evaluating 4 tasks. Thus, it takes $2 \times 20 \times 4 = 160$ times of evaluation. In the last round, there is only one 2-task merging and evaluation on 8 tasks. It takes $1 \times 20 \times 8 = 160$ times of evaluation. In total, NMMAP takes 480 times of evaluations. Generalizing the calculation, the number of rounds can be calculated by $\log N$. In each round, we need to evaluate $T \cdot N/2^i \cdot 2^i = TN$ where $T$ is the number of scaling coefficient vectors needed to be evaluated when the number of tasks is 2. In the above example, $T = 20$. Thus, the time complexity is $O(TN \log N)$. We rewrite it as $O(N \log N)$ if ignoring $T$ which is a constant.

### E.2.3 EXPLANATION OF CURSE OF DIMENSIONALITY

Figure 11 illustrates the curse of dimensionality through the discretizations of scaling coefficients for task-specific optimization. The left plot shows a 2D grid of scaling coefficients for two tasks, while the right plot extends this to three tasks (i.e., a 3D grid). In both cases, the space of possible scaling coefficients is discretized uniformly, with a resolution determined by the parameter `pts_per_dim`, which refers to the number of discretized points along each axis (dimension). The spacing between adjacent points, denoted as $\Delta$, is given by $\Delta = \frac{1}{\texttt{pts\_per\_dim}-1}$. As the number of dimensions (tasks) increases, the total number of grid points grows exponentially. Specifically, the number of grid points in $d$ dimensions is given by $(\texttt{pts\_per\_dim})^d$, leading to a combinatorial explosion in the search space. This exponential growth exemplifies the *curse of dimensionality*, rendering brute-force search methods computationally infeasible as the number of tasks (dimensions) increases beyond a small threshold. Therefore, while grid search may be adequate for low-dimensional problems, it quickly becomes impractical in higher-dimensional settings.

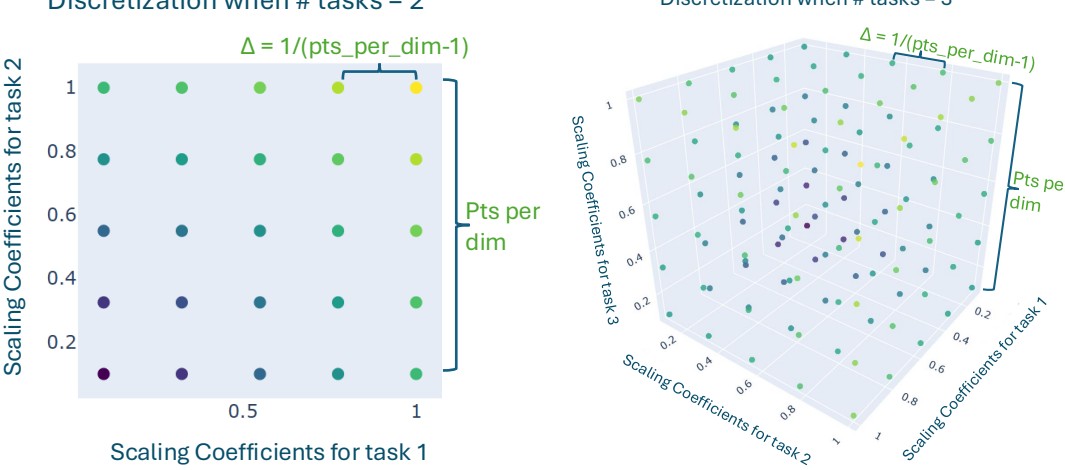

Figure 11: The discretization of scaling coefficients when the number of tasks is 2 and 3. As the dimension grows, preserving the same number of pts per dimension results in exponentially more grid points. This serves to illustrate why when the dimension is low (e.g., < 4), we can regard the brute-force direct search method as ground truth, but it becomes insufficient when the dimension is higher.

### E.2.4 INTERMEDIATE PARETO FRONTS IN EACH ROUND

To determine if NMMAP affects performance negatively when compared to merging multiple models in a single operation. We show the intermediate Pareto fronts obtained when merging 8 models using nested-merging MAP in Figure 12, where we merge two models at a time. The figures illustrate the intermediate Pareto fronts obtained, where $A\_B$ means the model obtained by merging model A and model B.

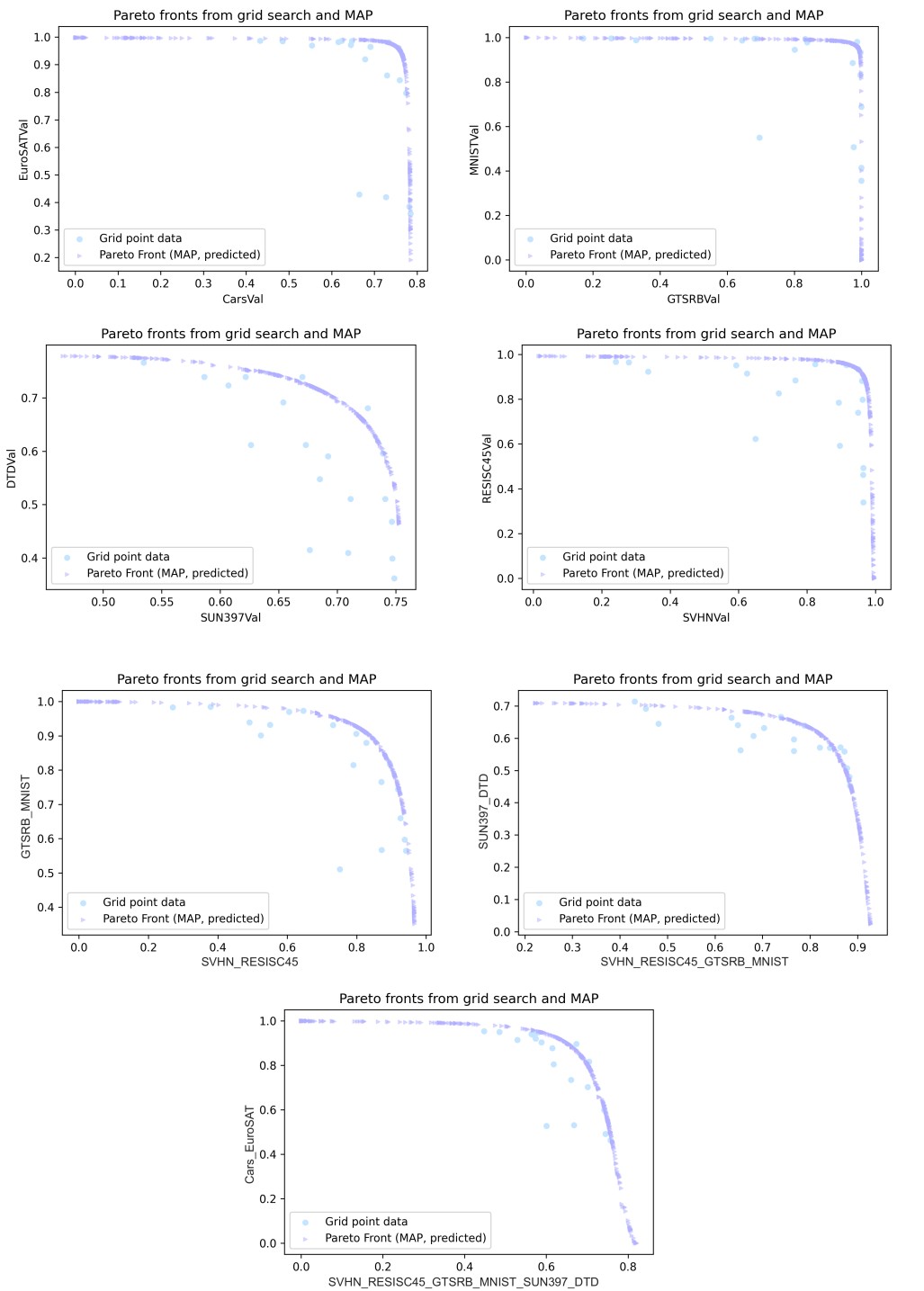

Figure 12: Illustration of the sequential steps involved in merging 8 models using nested-merging MAP (left to right, top to bottom). The figures show the intermediate Pareto fronts obtained, where $A\_B$ means the model obtained by merging model A and model B.

### E.2.5 RESULTS ON USING NESTED MERGING TO MERGE 8 MODELS

We further evaluate the effectiveness of nested-merging MAP. Supplemental Figure 12 shows the intermediate Pareto fronts obtained using nested-merging MAP (NMMAP) sequentially. In addition, we compared the performance of NMMAP with Plain MAP (Algorithm 1). The results are shown in Table 9. The results confirm that nested-merging MAP can obtain comparable results to those obtained by MAP while using much fewer evaluation runs. As discussed in Appendix E.2.2, in total, we run $40 \times 4 + 80 \times 2 + 160 \times 1 = 480$ evaluations. In contrast, when running the Plain MAP (Algorithm 1), we used 280 coefficients and evaluate $280 \times 8 = 2240$ times. In addition, while being suboptimal to MAP (Algorithm 1), nested-merging MAP can still outperform other baseline methods such as TA and DARE-TA in accommodating various user preferences with relatively low computational cost, especially when the number of tasks is high.

Table 9: We compared nested-merging MAP with MAP and baseline methods in merging 8 models using a set of 10 preference vectors in terms of preference-weighted sum of accuracies.

| Metric | Single-task models | Nested-merging MAP | Plain MAP | TIES-merging | DARE-TIES | Task Arithmetic (TA) | DARE-TA |
|---|---|---|---|---|---|---|---|
| Preference weighted sum ($\uparrow$) | 90.05±3.02 | 67.05±3.89 | **72.12**±3.78 | 70.79±3.81 | 70.51±3.58 | 59.44±5.68 | 63.14±4.91 |

### E.3 BAYESIAN MAP

We further improve the efficiency of MAP by proposing an adaptive Bayesian sampling algorithm. This Bayesian approach samples the points from regions with the highest level of uncertainty, where uncertainty is quantified with the Upper Confidence Bound (UCB). Please refer to Algorithm 3 for more details about the algorithm.

The discretization method shown in the Figure 13 illustrates how continuous decision variables $\mathbf{c}$ are divided (or "binned") based on their angular components, in either 2D polar or 3D spherical coordinate systems.

In general, a point $\mathbf{c} = (c_1, c_2, \ldots, c_n)$ in spherical coordinates is represented using:

$$\begin{cases} c_1 = r \cos \phi_1 \\ c_2 = r \sin \phi_1 \cos \phi_2 \\ c_3 = r \sin \phi_1 \sin \phi_2 \cos \phi_3 \\ \vdots \\ c_{n-1} = r \sin \phi_1 \cdots \sin \phi_{n-2} \cos \phi_{n-1} \\ c_n = r \sin \phi_1 \cdots \sin \phi_{n-2} \sin \phi_{n-1} \end{cases}$$

where $r = \|\mathbf{c}\|_2$ is the radial distance (magnitude), $\phi_1, \ldots, \phi_{n-2} \in [0, \pi]$, $\phi_{n-1} \in [0, 2\pi]$.

To discretize the **direction** (angular component) of a decision vector $\mathbf{c} \in \mathbb{R}^n$, we follow the steps below:

1. **Normalization:** Convert $\mathbf{c}$ to a unit vector:

$$\hat{\mathbf{c}} = \frac{\mathbf{c}}{\|\mathbf{c}\|_2}$$

2. **Conversion to Spherical Coordinates:** Represent the direction of $\mathbf{c}$ using $(n-1)$ spherical angles $(\phi_1, \phi_2, \ldots, \phi_{n-1})$. These can be computed recursively as follows:

$$\phi_1 = \arccos\left(\frac{c_1}{\sqrt{c_1^2 + c_2^2 + \cdots + c_n^2}}\right)$$

$$\phi_2 = \arccos\left(\frac{c_2}{\sqrt{c_2^2 + \cdots + c_n^2}} \middle/ \sin\phi_1\right)$$

$$\vdots$$

$$\phi_{n-2} = \arccos\left(\frac{c_{n-2}}{\sqrt{c_{n-2}^2 + c_{n-1}^2 + c_n^2}} \middle/ \prod_{j=1}^{n-3} \sin\phi_j\right)$$

$$\phi_{n-1} = \arctan\left(\frac{c_n}{c_{n-1}}\right)$$

3. **Discretization of Angular Space:** Divide each angle $\phi_i$ into $k_i$ uniform bins. For example, if $k_i = 4$, then:

$$\phi_i \in \left[\frac{j\pi}{2k_i}, \frac{(j+1)\pi}{2k_i}\right), \quad \text{for } j = 0, 1, \ldots, k_i - 1$$

where the angular domain is restricted to $[0, \pi/2]$ to ensure non-negativity in all coordinates.

4. **Bin Assignment:** The final bin index for $\mathbf{c}$ is a tuple of angular bin indices:

$$\text{bin}(\mathbf{c}) = (b_1, b_2, \ldots, b_{n-1}), \quad \text{where } b_i = \left\lfloor\frac{\phi_i}{\Delta\phi_i}\right\rfloor$$

and $\Delta\phi_i = \frac{\pi}{2k_i}$ denotes the bin width for $\phi_i$.

Each bin corresponds to a distinct angular sector, grouping together direction vectors with similar orientations, regardless of their magnitudes.

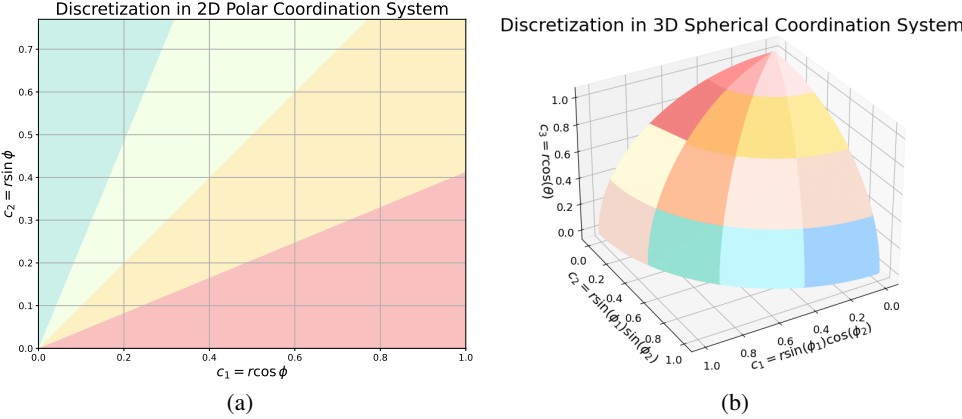

(a)  (b)

Figure 13: (a) Discretizing of two task scaling coefficients along the angular dimension in 2D polar coordinate system; (b) Discretizing of three task scaling coefficients along the angular dimensions in 3D spherical coordinate system.

---

**Algorithm 3** Bayesian Adaptive of Surrogate Model

---

**Input:** Number of iterations $J$, Buffer $\mathcal{B}$, Pretrained model $\boldsymbol{\theta}_{\text{pre}}$, Task vectors $\mathbf{v}_n$, Evaluators for task $N$, $M_n(\cdot)$, Discretization bin number $K$, sample size for every iteration $n_j$, $j = 0$ to $J$, Bootstrap dropping rate $\alpha = 20\%$, Bootstrap sampling number $Q = 30$.

$\mathcal{B} \leftarrow \emptyset$
**for** $j = 0$ *to* $J$ **do**
   **if** $j = 0$ **then**
      | Sample $n_0$ scaling coefficients $\{\mathbf{c}_i\}_{i=1}^{n_j}$ from $U([0,1]^N)$
   **else**
      | Sample $n_j$ scaling coefficients $\{\mathbf{c}_i\}_{i=1}^{n_j}$ based on the posterior distribution
   **end**
   **for** $i = 0$ *to* $n_j$ **do**
      | Merge the model $\theta_m(\mathbf{c}_i) = \boldsymbol{\theta}_{\text{pre}} + \mathbf{c}_i \cdot \mathbf{v}_n$ Evaluate $m_{n,i} = M_n(\theta_m(\mathbf{c}_i))$ $\mathcal{B} \leftarrow \mathcal{B} \cup \{(\mathbf{c}_i, m_{n,i})\}$
   **end**
   Fit the quadratic approximation surrogate model $\tilde{M}_n$ by learning $\mathbf{A}_n^*, \mathbf{b}_n^*, e_n^*$ in Equation (5).
   Discretize the scaling coefficients along the angular dimensions in hyper-spherical coordinates
    (see Figure 13 as examples)
   **for** $k = 0$ *to* $K$ **do**
      Calculate the mean of $L2$ loss between $\tilde{M}_n(\mathbf{c}_i)$ and $M_t(\mathbf{c}_i)$, where $\mathbf{c}_i$ are in bin $k$, denoted
        as $\text{mean}_k$
      Bootstrap to estimate the standard deviation of the losses.
      **for** $q = 0$ *to* $Q$ **do**
         | Randomly (uniformly) drop $\alpha$ scaling coefficient in bin $k$ Calculate the mean of $L2$ loss
         | between $\tilde{M}_n(\mathbf{c}_i)$ and $M_t(\mathbf{c}_i)$ with the rest points and denoted with $l_q$
      **end**
      Calculate the standard deviation of the $\{l_q\}_{q=0}^Q$ and denoted as $\text{std}_k$ $\text{score}_k = \text{mean}_k + \frac{1}{2}\text{std}_k$
   **end**
   Calculate probability distribution across the discretized bins by $\text{score}_k$ as the posterior sampling
    strategy in the next round
**end**

---

### E.3.1 PERFORMANCE OF BAYESIAN MAP

Please refer to Figure 14 for the experimental results comparing Bayesian MAP Algorithm 3 with MAP Algorithm 1. The very left column shows the names of the 2 tasks being merged. Below, we define points (pts) as scaling coefficients and evaluation metrics of the corresponding merged models. Please note that the results shown in this figure are the mean of 7 merging tasks that merge 2 models: DTD+Cars, DTD+RESISC45, EuroSAT+RESISC45, GTSRB+Cars, GTSRB+RESISC45, RESISC45+SVHN, SUN397+SVHN. The Pareto front estimated by Bayesian MAP with only one iteration (6+3 pts) is shown in Figure 15.

The experiments are initialized with 6 pairs of $\mathbf{c}$, $\{M_n\}_{n=1}^N$ (iter 0). In every following iteration, we sample more points following the Bayesian adaptive sampling algorithm. We compare the Bayesian adaptive sampling beginning with 6 points and adding 3 additional points ($6 + 3$ pts) with running Algorithm 1 with 9 points in a row. We also compare the Bayesian adaptive sampling beginning with 6 points and adding 3 additional points for 2 times ($6 + 2 \times 3$ pts) with running Algorithm 1 with 12 points in a row. We show that, in most cases, utilizing the same number of points, Bayesian adaptive sampling performs better than the run-in-a-row scheme in Algorithm 1.

In conclusion, when the number of data points (scaling coefficients and evaluation metrics of the corresponding merged models) is small, Bayesian MAP Algorithm 3 performs better than MAP Algorithm 1. As the number of data points increases, their performance becomes closer. Thus, we recommend implementing Bayesian MAP when computational resources are very limited and the number of models (tasks) to merge is not high.

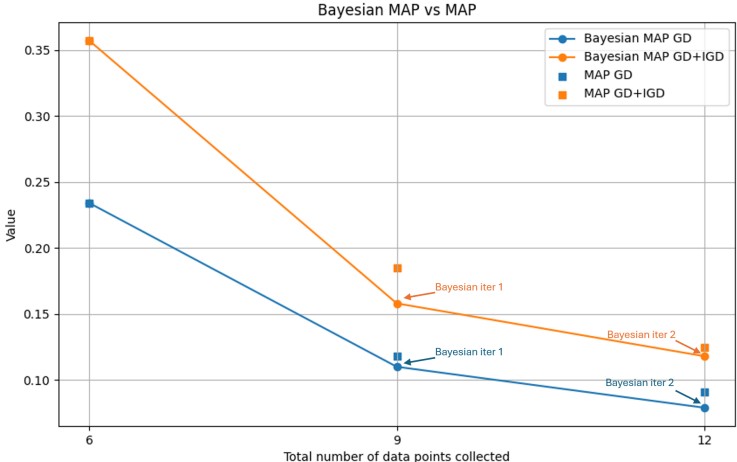

Figure 14: Bayesian MAP compared to MAP. The x-axis represents the number of points used by either MAP or Bayesian MAP, while the y-axis represents the value for IGD or GD. We compared MAP with 6, 9, and 12 points, and Bayesian MAP with 6 initial points, sampling 3 more points each round for two rounds.

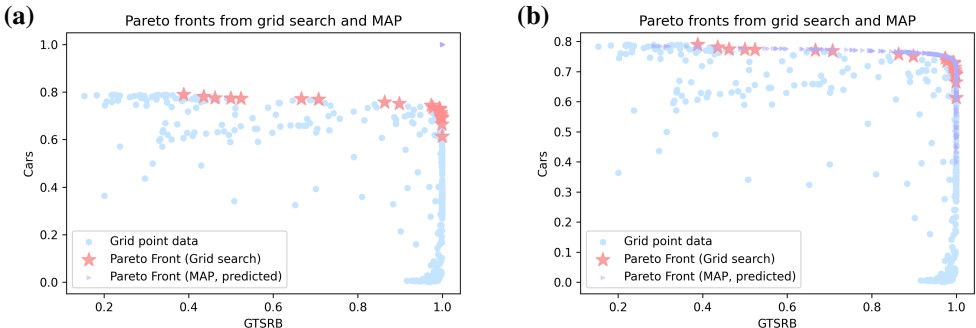

Figure 15: (a) This plot shows a failure case of the amortized Pareto front when we only have 6 initial randomly sampled pairs of $(\mathbf{c}, \{M_i(\theta_m(\mathbf{c}))\}_{i=1}^N)$ (b) After one iteration of Bayesian adaptive sampling for 3 more pairs (9 in total), the amortized Pareto front is much better than the initial Pareto front.

# F  POTENTIAL Q & A

In this section, we anticipate and address some potential questions that readers might have.

## F.1  THE AUTHORS ASSUME A SETTING WHERE IT IS COMPUTATIONALLY EXPENSIVE TO QUERY THE MODEL EVALUATIONS. BUT IS THIS THE REALITY?

Yes, to get the Pareto front of tasks that have trade-offs. We need to have a lot of data points to show the trade-off. Here, each data point is the evaluation metric of a model. To get the ground truth evaluation metric of a task (e.g. let's say classification), we need to run the evaluation script of the model to determine the metric. If we have 4 tasks, and we set 5 possible values (let's say 0.2, 0.4, 0.6, 0.8, 1) for the scaling coefficient vector of each model specialized for one task. We will have to evaluate $5^4 = 625$ models on all 4 tasks. Assuming each evaluation takes 1 minute, evaluating all the models on all the 4 tasks will take 625 models $\times$ 4 tasks $\times$ 1 minute = 2500 minutes = 41.7 hours which is expensive. In big O notation, assuming we have T tasks, the time of evaluation for each task is $T$. The time (computational) complexity is $O(TN2^N)$.

## F.2  HOW IS MAP DIFFERENT FROM OTHER MOOP METHODS, SUCH AS LINEAR SCALARIZATION?

**Computational Efficiency**  MAP is computationally efficient because it leverages a surrogate model (the quadratic approximation) to estimate the evaluation metrics for different model combinations. Once the quadratic model is learned, solving the optimization problem (approximating the Pareto front) is much faster because the complexity is reduced to optimizing over the quadratic approximation, rather than needing to directly evaluate the task metrics over the entire solution space.

Additionally, MAP does not require gradient descent or re-training when merging models and the surrogate model provides an efficient way to approximate the Pareto front without re-evaluating every model configuration.

On the other hand, linear scalarization involves computing a weighted sum of the objectives and solving it as a single-objective optimization problem. However, it often requires multiple optimization runs with different weight configurations to explore different points on the Pareto front. Since linear scalarization does not model non-linear relationships, it might require more iterations or grid searches over different weights to achieve a decent approximation of the Pareto front. This can become computationally expensive, especially for a large number of tasks or highly complex models.

**Non-convex Pareto Fronts**  Many real-world multi-objective optimization problems have nonconvex Pareto fronts, which require more sophisticated methods. While linear scalarization can only explore the convex regions of the Pareto front, NSGA-III allows MAP to discover nonconvex regions of the Pareto front. It does this by evolving a population of solutions that are non-dominated (i.e., not Pareto dominated by any other solution) and spreading these solutions across the entire Pareto front, including both convex and non-convex regions.

## F.3  WHY DOES QUADRATIC APPROXIMATION HAVE A LOWER COST?

For the MAP algorithm (Algorithm 1), the time complexity is the same as what we mentioned above; what is different is that we fitted an approximation of the evaluation metrics. We only need the scaling coefficient vectors to input to a quadratic function model to be able to get the estimated evaluation score. Running the quadratic function once takes only $3.91 \times 10^{-6}s \pm 894 \times 10^{-9}s$. Thus, evaluating 2500 times takes $< 2500 \times 4 \times 10^{-6}s = 10^{-2}s$.

## F.4  WHY NOT COMPARE TO GIT-REBASIN (AINSWORTH ET AL., 2023)?

We did not compare our method to Git Re-Basin because their study focuses on the scenarios of merging the models from different initializations, whereas our background works on the same initialization of different fine-tuned models.

## F.5 WHY DOES NESTED-MERGING MAP NOT GIVE A COMPLETE PARETO FRONT?

In nested-merging MAP, we merge the models in pairs. For example, there are $model_i$ for task i as individual fine-tuned models where $i = 1, 2, 3, 4$. We first merge model 1 and model 2 given the preference of the user between task 1 and task 2. We then get $model_{1,2}$. At the same time, we merge model 3 and model 4 given the preference of the user between task 3 and task 4 and get $model_{3,4}$. Finally, we merge the $model_{1,2}$ and $model_{3,4}$ given the preference of user to get $model_{1,2,3,4}$. We output the $model_{1,2,3,4}$ as the output merged model of the algorithm. Please note that the merging order in the algorithm should be decided by the loss function clustering. It is a heuristic decision and does not always dominate other merging orders. Thus, we choose not to emphasize the contribution of this order. The practitioner may use any order they think can be helpful for merging.

## F.6 IT SEEMS NESTED-MERGING MAP PERFORMS WORSE THAN MAP. WHAT IS THE MEANING OF IT?

In theory, the surrogate model can be easily fitted when the number of tasks is low. When it is high (e.g. 8), the fit of the surrogate model can no longer be a near-perfect approximation (please find the $R^2$ in Table 3). Thus, even if the NMMAP can only find the suboptimal solution, it is still comparable to MAP Algorithm 1 according to Table 9. We understand in general that NMMAP does not find the global optimum, but please kindly keep in mind that even if their performance is comparable, NMMAP takes way less computation for evaluation.

## F.7 WHY DOES NESTED-MERGING MAP NOT OUTPUT THE OPTIMAL SOLUTION?

The solution found by nested-merging MAP (NMMAP) is indeed suboptimal given the limited search space compared with merging all models at once. However, it does not mean that it is not useful in all situations. When the number of tasks is high, it has comparable performance to MAP while consuming much fewer computations for evaluation.

## F.8 HOW DO YOU DEAL WITH GRADIENT AND HESSIAN IN THE SECOND-ORDER TAYLOR EXPANSION?

Notations:

- $p$ as the number of parameters in the pre-trained model (also the number of parameters in each task vector).
- $\mathbf{V}$ is the matrix of task vectors of different N tasks. Thus, $\mathbf{V} \in \mathbb{R}^{p \times N}$.
- $\mathbf{c}$ is the scaling coefficient vector $\in \mathbb{R}^N$.
- $M_n$ is the metric (e.g. accuracy) for the task n.

$$M_n(\mathbf{c}) = \underbrace{M_n(\boldsymbol{\theta}_{\text{pre}})}_{\in \mathbb{R}} + \underbrace{\nabla M_n(\boldsymbol{\theta}_{\text{pre}})^\top}_{\in \mathbb{R}^{1 \times p}} \underbrace{\mathbf{V}}_{\in \mathbb{R}^{p \times N}} \underbrace{\mathbf{c}}_{\in \mathbb{R}^{N \times 1}} + \frac{1}{2} \underbrace{(\mathbf{Vc})^\top}_{\in \mathbb{R}^{1 \times p}} \underbrace{\mathbf{H}_n(\boldsymbol{\theta}_{\text{pre}})}_{\in \mathbb{R}^{p \times p}} \underbrace{\mathbf{Vc}}_{\in \mathbb{R}^{p \times 1}} + \underbrace{R_n}_{\in \mathbb{R}} \tag{6}$$

$$= \underbrace{e_n}_{\in \mathbb{R}} + \underbrace{\mathbf{b}_n^\top}_{\in \mathbb{R}^{1 \times N}} \underbrace{\mathbf{c}}_{\in \mathbb{R}^{N \times 1}} + \underbrace{\mathbf{c}^\top}_{\in \mathbb{R}^{1 \times N}} \underbrace{\mathbf{A}_n}_{\in \mathbb{R}^{N \times N}} \underbrace{\mathbf{c}}_{\in \mathbb{R}^{N \times 1}} \tag{7}$$

$$\tag{8}$$

$$\tilde{M}_n(\mathbf{c}; \mathbf{A}_n, \mathbf{b}_n, e_n) \equiv e_n + \mathbf{b}_n^\top \mathbf{c} + \frac{1}{2} \mathbf{c}^\top \mathbf{A}_n \mathbf{c}$$

where

$$\mathbf{A}_n = \mathbf{V}^\top \mathbf{H}_n(\boldsymbol{\theta}_{\text{pre}}) \mathbf{V} \in \mathbb{R}^{N \times N}, \mathbf{b}_n = \mathbf{V}^\top \nabla M_n(\boldsymbol{\theta}_{\text{pre}}) \in \mathbb{R}^N, e_n = M_n(\boldsymbol{\theta}_{\text{pre}}) + R_n \tag{9}$$

Please notice that $\mathbf{A}$ is a symmetric matrix. Specifically, when the number of tasks is 2, we have:

$$\tilde{M}_1(\mathbf{c}; \mathbf{A}_1, \mathbf{b}_1, e_1) \equiv e_1 + \mathbf{b}_1^\top \mathbf{c} + \frac{1}{2}\mathbf{c}^\top \mathbf{A}_1 \mathbf{c} \tag{10}$$

$$= \frac{1}{2}A_{1,11}c_1^2 + A_{1,12}c_1 c_2 + \frac{1}{2}A_{1,22}c_2^2 + b_{1,1}c_1 + b_{1,2}c_2 + e_1 \tag{11}$$

$$\tilde{M}_2(\mathbf{c}; \mathbf{A}_2, \mathbf{b}_2, e_2) \equiv e_2 + \mathbf{b}_2^\top \mathbf{c} + \frac{1}{2}\mathbf{c}^\top \mathbf{A}_2 \mathbf{c} \tag{12}$$

$$= \frac{1}{2}A_{2,11}c_1^2 + A_{2,12}c_1 c_2 + \frac{1}{2}A_{2,22}c_2^2 + b_{2,1}c_1 + b_{2,2}c_2 + e_2 \tag{13}$$

$$\tag{14}$$

We don't calculate the gradient or Hessian to get $\mathbf{A}_1$, $\mathbf{A}_2$, $\mathbf{b}_1$, $\mathbf{b}_2$, $e_1$ and $e_2$. We use linear regression to estimate them. How? Given a $(c_1, c_2)$ pair, we can define a merged model $\boldsymbol{\theta}_{\mathrm{merge}}(c_1, c_2)$. We evaluate the merged model on task 1 and task 2 to get the metrics (e.g. accuracy). Given 20 pairs of $(c_1, c_2)$, we would be able to evaluate and get 20 corresponding, $(M_1, M_2)$ which are metrics for task 1 and task 2. Thus, we can fit the surrogate model $\tilde{M}_1(\mathbf{c}; \mathbf{A}_1, \mathbf{b}_1, e_1)$ and $\tilde{M}_2(\mathbf{c}; \mathbf{A}_2, \mathbf{b}_2, e_2)$.

