# OpenReview forum: "MAP: Low-compute Model Merging with Amortized Pareto Fronts via Quadratic Approximation"
_ICLR.cc/2025/Conference — ICLR 2025 Poster_

### Official Review · Reviewer_9bUH · 2024-10-31

**Soundness:** 3
**Presentation:** 3
**Contribution:** 2
**Rating:** 5
**Confidence:** 4

**Summary:**

The paper introduces Model Merging with Amortized Pareto Front (MAP), a low-compute algorithm that merges multiple single-task models into a multitask model by efficiently identifying a Pareto set of scaling coefficients. MAP uses quadratic surrogate models to reduce evaluation costs while providing flexible solutions to balance competing task objectives.

**Strengths:**

1. This is the first algorithm to estimate the Pareto front for task-vector-based model merging without relying on gradient descent, which is often computationally expensive.

2. The Nested MAP variant reduces computational complexity, making it suitable for large-scale problems.

**Weaknesses:**

1. The motivation for applying Multi-Objective Optimization Problems (MOOP) in model merging needs further clarification. While this work represents a direct application of MOOP in this area, it lacks an in-depth explanation of why MOOP would be advantageous over traditional gradient descent-based methods.

2. To enhance the clarity and impact of the paper, consider including a direct comparison with gradient descent-based optimization. Specifically, the authors could discuss MOOP’s potential benefits in terms of computational efficiency, ability to handle non-differentiable objectives, flexibility in exploring trade-offs, and its capacity to fully explore the Pareto front, which gradient-based methods may not achieve. This comparison would help elucidate the unique value of MOOP for model merging.

3, The paper would benefit from a more thorough comparative analysis with recent relevant works, particularly "Knowledge Fusion by Evolving Weights of Language Models" and "It's Morphing Time: Unleashing the Potential of Multiple LLMs via Multi-objective Optimization." Both studies propose innovative methods for model merging with evolutionary approaches.
A direct comparison with these methods could clarify the specific advancements and trade-offs associated with the MAP approach, such as variations in fusion strategies, optimization techniques, or performance across diverse benchmarks. Discussing how MAP aligns or diverges in terms of methodology, effectiveness, or scope will provide readers with a more complete understanding of its contribution to the field.

**Questions:**

1. The paper lacks comparative experiments with other established MOOP algorithms, such as MOEA/D and NSGA-II. Including these comparisons would enhance the evaluation of both solution quality and computational efficiency, providing a clearer context for assessing the performance of the proposed method. Additionally, brute force may not be the most appropriate baseline for this type of analysis and could be replaced by a simple MOOP method like MOEA/D.

2. The experiments related to large language models are somewhat limited. Typically, mainstream model fusion effectiveness is tested on benchmarks like math and code tasks, as seen in recent work such as DARE (arXiv: 2311.03099). Including comparisons on these types of benchmarks would lend stronger support to the method’s effectiveness relative to established model fusion approaches.

3. The paper would benefit from additional results or comparative analysis with other state-of-the-art model merging methods, such as Adamerging (ICLR 2024) and DELLA-Merging (arXiv: 2406.11617v1). Adding these would help situate the proposed method within the current landscape and highlight any unique strengths or trade-offs.

---

> ### Author Response · Authors · 2024-11-21
>
> Thanks for the valuable suggestions and questions.
>
> The reviewer 6MqJ proposed that MAP is already taken by Maximum a posteriori. Thus, we changed our paper name to LocMAP. In the rebuttal, we kept using MAP but, in the manuscript, due to consistency, we already changed all instances to LocMAP.
>
> ## Why MOOP-based method is better than gradient descent-based methods
> W1: The motivation for applying Multi-Objective Optimization Problems (MOOP) in model merging needs further clarification. While this work represents a direct application of MOOP in this area, it lacks an in-depth explanation of why MOOP would be advantageous over traditional gradient descent-based methods. To enhance the clarity and impact of the paper, consider including a direct comparison with gradient descent-based optimization. Specifically, the authors could discuss MOOP’s potential benefits in terms of computational efficiency, ability to handle non-differentiable objectives, flexibility in exploring trade-offs, and its capacity to fully explore the Pareto front, which gradient-based methods may not achieve. This comparison would help elucidate the unique value of MOOP for model merging.
>
> A1: Thanks for the suggestion!
>
> - [Computational Efficiency] Gradient-based methods exactly does have its advantages on performance. In contrast, model-merging based method is cheaper to computes. Especially in CPU-only scenarios.
> - [Avoiding Retraining for Different Preference Weighting] The goal of MAP is to build a Pareto fronts for model-merging facing all kinds of preference. Once the Pareto front is estimated, there are no calculation need to adapt any preference. On the other side, gradient decent-based method need to redo training given different preference vectors because the target function has been changed.
> - [Capture Trade-offs]: Since our algorithm explicitly seeks Pareto optimality, it is helpful for practitioners to understand the trade-offs between tasks. Gradient descent methods may not effectively explore the trade-offs between tasks because they often optimize a (preference) weighted sum of losses, which doesn't guarantee a Pareto optimal solution.
>
> We added these discussions to our paper as well. Thank you!
>
> ##
> W2: The paper would benefit from a more thorough comparative analysis with recent relevant works, particularly "Knowledge Fusion by Evolving Weights of Language Models" and "It's Morphing Time: Unleashing the Potential of Multiple LLMs via Multi-objective Optimization." Both studies propose innovative methods for model merging with evolutionary approaches. A direct comparison with these methods could clarify the specific advancements and trade-offs associated with the MAP approach, such as variations in fusion strategies, optimization techniques, or performance across diverse benchmarks. Discussing how MAP aligns or diverges in terms of methodology, effectiveness, or scope will provide readers with a more complete understanding of its contribution to the field.
>
> A2:
> Thanks for mentioning these 2 related work. We have included them in our related work section.
>
> ### Variations in fusion strategies:
> - MAP: Our method amortizes the computational cost by fitting surrogate models using a limited set of scaling coefficients. Once the quadratic surrogate models are established, MAP applies multi-objective optimization algorithms to identify Pareto-optimal solutions.
> - Knowledge Fusion by Evolving Weight: This method relies on "mutation" and crossover operations typical of evolutionary algorithms, exploring the parameter space by combining and perturbing existing solutions. Models are evaluated on development datasets, and only those that improve upon their predecessors are retained, guiding the population toward better-performing solutions.
> - It's Morphing Time (MM-MO): this method employs Bayesian optimization with a weak-to-strong approach and utilizes Fisher information to improve the selection of configurations for evaluation, aiming to find optimal merging configurations within limited computational budgets.
>
> ### The alignment between the 3 methods:
> - All three methods are based on model merging
> - Both MAP and MM-MO formulate the problem as a multi-objective optimization task, seeking to balance performance across different objectives.
> - MAP and the Knowledge Fusion method both emphasize merging models without the need for extra training data or extensive retraining.
>
> ### The divergence between the 3 methods:
> - MAP is designed for low-compute environments, focusing on efficiency. The surrogate model we used is the quadratic model.
> - Knowledge Fusion may require more computational resources due to the population-based evolutionary process.
> - MM-MO employs black-box Bayesian optimization with enhanced acquisition strategies which requires multiple rounds of updating. The surrogate model it used is Gaussian process.
>
> (To be continued)

---

> ### Author Response · Authors · 2024-11-21
>
> ## Baseline method
> Q1: The paper lacks comparative experiments with other established MOOP algorithms, such as MOEA/D and NSGA-II. Including these comparisons would enhance the evaluation of both solution quality and computational efficiency, providing a clearer context for assessing the performance of the proposed method. Additionally, brute force may not be the most appropriate baseline for this type of analysis and could be replaced by a simple MOOP method like MOEA/D.
>
> A1: Thank you for raising this interesting point! We initially opted to use a brute-force method as a baseline instead of an MOOP algorithm due to limited computational resources at the time. This approach allowed us to pre-select scaling coefficients and store the corresponding evaluation results for reuse. Now that the brute-force experiments are complete, we shall extend our work to include more experiments with evolutionary algorithms, such as MOEA/D.
>
> Below we include the MOEA/D baseline experiments for 2, 3, ...,  8 tasks and have computed the win rate between MOEA/D and MAP below. Due to computation limit, we used population size of 50 and number of generations of 25. For dimension 8, this results in 50 * 25 * 8 = 10,000 evaluations. On the other hand, MAP used 250 * 8 = 2000 evaluations. We include the results and our discussions below as well as updating in the manuscript.
>
> | N | MAP Win rate (MAP vs MOEA/D)| MAP Win rate (MAP vs Direct Search)| # evaluations (MOEA/D)|  # evaluations (MAP)|# evaluations (Direct Search)|
> |-----------|------------|------------|------------|------------|------------|
> |  2          | 51.0% $\pm$ 0.02 |49.81% $\pm$ 0.30 | 2500  | 60 | 400 |
> |  3          | 58.6%  $\pm$ 0.01 | 46.90% $\pm$ 0.71 | 3750  | 150 | 900 |
> |  4         | 51.3%  $\pm$ 0.02 | 50.67% $\pm$ 2.44 | 5000  | 240 | 1200 |
> |  5          | 50.6%  $\pm$ 0.02 | 53.00% $\pm$ 1.88 | 6250  | 425 | 2500 |
> |  6          | 52.2%  $\pm$ 0.02 | 60.71% $\pm$ 1.34 | 7500  | 600 | 3000 |
> |  7          | 51.6%  $\pm$ 0.02 | 63.42% $\pm$ 1.91 | 8750  | 980 | 7000 |
> |  8          | 53.2% $\pm$  0.07 | 65.58% $\pm$ 0.94 |10000 | 2000 | 8000 |
>
> Below are the discussions to the results above.
>
> #### Pareto Front Diversity
> In our experiments, MOEA/D struggled to achieve a diverse Pareto front, as can be seen from updated manuscript Figure 5. The solutions found by MOEA/D exhibited clustering and a lack of adequate spread across the Pareto front. In contrast, MAP consistently produced a well-distributed Pareto front.
>
> #### Computational Efficiency
> The computational cost of MOEA/D was significantly higher than that of MAP. For example, with a population size of 50 and 20 generations (which produced the best results for MOEA/D in terms of diversity), the total number of evaluations amounted to 2500 for two tasks. In comparison, MAP achieved its results with only 60 total evaluations—a reduction of over 97%.
>
> #### Hyperparameter Considerations
> The population size and number of generations are hyperparameters that can significantly influence the performance of MOEA/D. Due to computational constraints, we limited our experiments to configurations with population sizes of up to 50 and generations up to 25 (totaling around 25 * 50 * 2 = 2500 evaluations). These choices were made to ensure the experiments were feasible while providing a reasonable baseline for comparison. We acknowledge that further tuning of these hyperparameters (e.g., larger population sizes or generations) could potentially improve MOEA/D’s performance, but such exploration was beyond the scope of our current computational resources, further demonstrating the low compute nature of MAP.

---

> ### Author Response · Authors · 2024-11-22
>
> ## Additional exps on LLMs
> Q2: The experiments related to large language models are somewhat limited. Typically, mainstream model fusion effectiveness is tested on benchmarks like math and code tasks, as seen in recent work such as DARE (arXiv: 2311.03099). Including comparisons on these types of benchmarks would lend stronger support to the method’s effectiveness relative to established model fusion approaches.
>
> A2: Thank you for the suggestion! Merging a coding LLM with a Math LLM is indeed an intriguing experiment, and we’re excited to explore it further.
>
> The results of our experiments merging Math LLM and Coding LLM are presented in Table 7 (Page 23) and Figure 9 (Page 24) of the updated manuscript. Both the table and the figure demonstrate that MAP has been highly effective in approximating the Pareto fronts.
> | Task Pair       | GD               | IGD              | GD+IGD             |
> |---------------------|------------------|------------------|--------------------|
> | Math + Code         | $0.039_{0.009}$ | $0.018_{0.002}$  | $0.057_{0.008}$    |
>
> ## Additional baselines
> Q3: The paper would benefit from additional results or comparative analysis with other state-of-the-art model merging methods, such as Adamerging (ICLR 2024) and DELLA-Merging (arXiv: 2406.11617v1). Adding these would help situate the proposed method within the current landscape and highlight any unique strengths or trade-offs.
>
> A3: We have updated the manuscript to include the results for Adamerging++ and DELLA-Merging. Specifically, in Figure 5, we illustrate the performance of Adamerging++ and DELLA-Merging as single data points on the Pareto fronts for cases where the number of tasks equals 2. For scenarios involving more than 2 tasks, we present a comparative evaluation in Table 4. We sample a set of 20 normalized preference vectors and computing the preference-weighted sum of accuracies for both Adamerging++ and DELLA-Merging.
>
> | # tasks                      | 2                | 3                | 4                | 5                | 6                | 7                | 8                |
> |------------------------------|------------------|------------------|------------------|------------------|------------------|------------------|------------------|
> | Single task models           | 75.84±1.76       | 77.03±1.84       | 82.43±4.40       | 87.69±4.50       | 88.52±4.02       | 89.26±3.58       | 90.62±2.52       |
> | MTL                          | 73.63±0.30       | 75.13±1.00       | 80.10±2.79       | 84.93±3.58       | 86.78±2.94       | 87.40±2.56       | 89.11±2.36       |
> | Model soups                  | 67.79±1.46       | 64.25±2.15       | 66.04±3.22       | 67.01±3.42       | 63.11±1.99       | 63.35±2.17       | 64.36±2.77       |
> | TIES-merging                 | 69.30±0.33       | 67.60±0.58       | 71.79±2.93       | 76.49±3.10       | 73.74±2.96       | 72.54±2.87       | 72.24±1.91       |
> | DARE-TIES                    | 67.62±1.65       | 66.49±2.34       | 71.39±4.45       | 74.55±4.55       | 73.34±4.10       | 71.43±3.84       | 71.89±2.86       |
> | Task Arithmetic              | **70.73±1.84**   | 61.15±2.33       | 52.69±4.23       | 61.58±4.62       | 51.37±3.84       | 39.79±3.97       | 60.77±2.84       |
> | TA with preference as weights| 69.22±1.40       | 66.88±2.37       | 68.73±5.48       | 71.92±5.50       | 68.13±4.69       | 68.14±4.20       | 68.17±2.89       |
> | DARE-TA                      | 70.61±0.22       | 64.18±1.24       | 58.04±8.19       | 65.39±7.03       | 56.76±7.01       | 46.75±5.73       | 64.51±3.81       |
> | Ada-Merging++                | 67.27±1.92       | 67.13±1.92       | 71.19±4.43       | 76.84±4.71       | 74.13±4.07       | 72.58±4.16       | 72.55±2.83       |
> | DELLA-Merging                | 67.10±2.08       | 65.92±2.48       | 70.71±4.31       | 74.43±4.32       | 72.64±3.77       | 71.16±3.95       | 71.49±2.83       |
> | MOEA/D                       | 70.22±1.46       | 67.94±1.79       | 70.85±4.58       | 72.03±4.03       | 67.88±3.09       | 69.06±2.97       | 68.59±2.89       |
> | **LocMAP**                   | 70.7±1.76        | **69.05±1.84**   | **72.84±4.4**    | **77.31±4.5**    | **74.26±4.02**   | **73.40±3.58**   | **72.96±2.52**   |

---

> > ### Author Response · Authors · 2024-11-29
> > **Follow-up on Rebuttal and Review Feedback**
> >
> > Dear reviewer 9bUH,
> >
> > We sincerely appreciate the time and efforts you've devoted to reviewing our work. We understand that your schedule may be quite busy. As the authors-reviewer discussion period draws to a close, we kindly request your attention to our responses. Our aim is to gain insights into whether our responses effectively address your concerns and to ascertain if there are any additional questions or points you would like to discuss. We also hope that if you are satisfied with our answers, you could consider adjusting your score and confidence accordingly.
> >
> > We look forward to the opportunity for further discussion with you. Thank you again very much for your thoughtful consideration.
> >
> > Best regards,
> > The Authors

---

> > > ### Author Response · Authors · 2024-12-02
> > >
> > > Dear reviewer 9bUH,
> > >
> > > We hope this message finds you well. We appreciate your reviews and comments. Your insights and feedback are invaluable to us, and we deeply appreciate the time and effort you dedicate to this process.
> > >
> > > If there’s anything we can clarify or assist with to make the review smoother, please kindly let us know. We are here to help in any way that might be needed.
> > >
> > > Thank you once again for your thoughtful consideration and support.
> > >
> > > Best regards,
> > >
> > > The Authors

---

### Official Review · Reviewer_97hV · 2024-11-01

**Soundness:** 3
**Presentation:** 2
**Contribution:** 3
**Rating:** 6
**Confidence:** 4

**Summary:**

The paper introduces a new model merging method that aims to approximate the pareto front of the performance of various model merging scaling factors by a quadratic approximation of the metric that is used for performance evaluation.
By not requiring a full search over possible scaling factors, the amount of computation that is needed is drastically reduced.
The authors show that this approach is favorable especially for a larger number of tasks, where the number of possible scaling factor combinations increases exponentially.

**Strengths:**

- The idea of focusing on trade-offs between tasks for which models are merged and Pareto fronts instead of only a single model merging solution is interesting and a useful reframing of model merging.

- The method is derived from sound theory and can reduce the cost of model merging.

- The method can be used as a plug-in addition to Task-Arithmetic-based merging schemes.

**Weaknesses:**

- The paper overall is not easy to follow. Many details are left to the reader and there is not always a clear flow in writing, requiring the reader to jump back-and-forth. In particular, the following points could be improved:

- Section 2.3: it is not immediately clear why these norms are calculated, because the fact that the method uses taylor approximations is only introduced at the end of it but even then it is unclear how it ties in with the bigger picture, esp. how closeness of parameters may be related to a taylor approximation of the evaluation metric. This could be clarified. In particular, it is directly showing empirical evidence that Assumption 1 may be valid but this only comes in the section afterwards.

- Section 3.1 (the main description of the method) is not written very well. For example, case 2 and 3: why is the "warping" by a sigmoid benificial and why does a softplus help in Case 3? Many details are left for the reader to figure out. Also, it is mentioned that you optimize Eq.5 in L252 but that you do it with gradient descent is loosely thrown in in L283. Overall, Eq.5 could be discussed more, too.

- The nested MAP (nMAP) is only described in Fig. 4 of the main paper and I can not seem to find any description of bMAP at all. Could you please clarify this? While I agree that how nested merging is done is very intuitive a better description would be helpful.

- It would be helpful to discuss related works more, in particular, Rame et al. 2023, who also seek to use Task-Arithmetic-based merging for Pareto fronts of multiple objectives

**Questions:**

- In Fig 2 it is not immediately clear to me why the brute force approach of finding the best multitask scaling factor performs worst, also since you call it gold standard. Could you please explain this a bit further? What does the direct search look for exactly? Is it just over Task Arithmetic scaling factors, and if so, what grid is used?

---

> ### Author Response · Authors · 2024-11-21
>
> Thank you for the great reviews! We appreciate your detailed examination of the paper. Below are the answers to the questions.
>
> Reviewer 6MqJ proposed that MAP is already taken by Maximum a posteriori. Thus, we changed our paper name to LocMAP. In the rebuttal, we kept using MAP but, in the manuscript, due to consistency, we already changed all instances to LocMAP.
>
> ## Remove the motivation section
>
> Q1: Section 2.3: it is not immediately clear why these norms are calculated, because the fact that the method uses Taylor approximations is only introduced at the end of it, but even then it is unclear how it ties in with the bigger picture, especially how the closeness of parameters may be related to a Taylor approximation of the evaluation metric. This could be clarified. In particular, it is directly showing empirical evidence that Assumption 1 may be valid, but this only comes in the section afterward.
>
> A1: Thanks for pointing this out. We agree that putting the motivation in section 2.3 is likely to confuse people. Therefore, we moved this part to section 3 along with the Taylor expansion in the updated manuscript.
>
> ## More discussion on nonlinear warper
>
> Q2: Section 3.1 (the main description of the method) is not written very well. For example, in cases 2 and 3: why is the "warping" by a sigmoid beneficial and why does a softplus help in Case 3? Many details are left for the reader to figure out. Also, it is mentioned that you optimize Eq. 5 in L252, but that you do it with gradient descent is loosely thrown in at L283. Overall, Eq. 5 could be discussed more, too.
>
> A2: Thanks for pointing this out. The "warping" by a sigmoid/softplus is beneficial because we want to restrict the estimation of the metric within the feasible solution space (as we pointed out in L267 and L270). We added more descriptions to further clarify this in the updated manuscript. Apart from that, we also added more discussion for equation 5 around L282. Thank you!
>
> ## More details in Nested MAP and Bayesian MAP
>
> Q3: The nested MAP (NMAP) is only described in Fig. 4 of the main paper and I cannot seem to find any description of NMAP at all. Could you please clarify this? While I agree that how nested merging is done is very intuitive, a better description would be helpful.
>
> A3: Because of space limitations, we moved the detailed description of BMAP to appendix E.3 and the detailed description of NMAP to appendix E.2. However, as reviewer 6MqJ suggested, we moved figure 5 to the appendix and thus, we could mention more of BMAP and NMAP in the main content (section 3.3). Please kindly refer to the updated manuscript.
>
> ## More related works
>
> Q4: It would be helpful to discuss related works more, in particular, Rame et al. 2023, who also seek to use Task-Arithmetic-based merging for Pareto fronts of multiple objectives.
>
> A4: Thanks for pointing that out! We included multiple papers:
>
> - [We have this in the original version] Ramé, Alexandre, et al. "Rewarded soups: towards Pareto-optimal alignment by interpolating weights fine-tuned on diverse rewards." Advances in Neural Information Processing Systems 36 (2024).
>
> - [Newly added] Ramé, Alexandre, et al. "Warp: On the benefits of weight-averaged rewarded policies." arXiv preprint arXiv:2406.16768 (2024).
>
> - [Newly added] Ramé, Alexandre, et al. "Warm: On the benefits of weight-averaged reward models." arXiv preprint arXiv:2401.12187 (2024).
>
> They are mentioned in Appendix Section B "MORE DISCUSSION ON RELATED WORK" in the updated manuscript.
>
> ## More explanation of direct search and grids
>
> Q5: In Fig. 2 it is not immediately clear to me why the brute force approach of finding the best multitask scaling factor performs worst, also since you call it the gold standard. Could you please explain this a bit further? What does the direct search look for exactly? Is it just over Task Arithmetic scaling factors, and if so, what grid is used?
>
> A5: Thanks for the valuable question.
>
> Yes, the direct search looks for Task Arithmetic scaling factors. The visualization of grid search was in Figure 5 and now is moved to Appendix Figure 10.
>
> As we mentioned in Tables 2 and 3, we only regard brute force direct search as the gold standard when the number of tasks N = 2, 3. When N > 3, due to the restrictions of computation we have, and also due to the curse of dimensionality, we cannot cover the search space in a fine-grained manner.
>
> In detail, when N = 2, we searched 200 grid points (takes 200 × 2 = 400 evaluations); when N = 3, we searched 300 grid points (#Eval = 300 × 3 = 900); when N = 4, we searched 300 grid points (#Eval = 300 × 4 = 1200); when N = 5, we searched 500 grid points (#Eval = 500 × 5 = 2500); when N = 6, we searched 500 grid points (#Eval = 500 × 6 = 3000); when N = 7, we searched 1000 grid points (#Eval = 1000 × 7 = 7000); when N = 8, we searched 1000 grid points (#Eval = 1000 × 8 = 8000).

---

> > ### Author Response · Authors · 2024-11-29
> > **Follow-up on Rebuttal and Review Feedback**
> >
> > Dear reviewer 97hV,
> >
> > We sincerely appreciate the time and efforts you've devoted to reviewing our work. We understand that your schedule may be quite busy. As the authors-reviewer discussion period draws to a close, we kindly request your attention to our responses. Our aim is to gain insights into whether our responses effectively address your concerns and to ascertain if there are any additional questions or points you would like to discuss. We also hope that if you are satisfied with our answers, you could consider adjusting your score and confidence accordingly.
> >
> > We look forward to the opportunity for further discussion with you. Thank you again very much for your thoughtful consideration.
> >
> > Best regards,
> > The Authors

---

> > > ### Author Response · Authors · 2024-12-02
> > >
> > > Dear reviewer 97hV,
> > >
> > > We hope you’re doing well. We’re reaching out with a gentle reminder that today marks the deadline to provide a response to our rebuttal. We truly value your feedback and the time you’ve invested in this process.
> > >
> > > Please kindly let us know if there’s anything we can clarify or assist with to help finalize the review.
> > >
> > > Thank you again for your time and effort!
> > >
> > > Best regards,
> > >
> > > The Authors

---

> > > > ### Comment · Reviewer_97hV · 2024-12-02
> > > > **Thank you for your response!**
> > > >
> > > > Dear Authors,
> > > >
> > > > Thank you very much for your extensive response and revisions.
> > > > I find the updated manuscript much easier to read and follow! For now this addresses my questions.
> > > > However, i'd like to ask the authors to continue improving some of the readability and formatting of the paper.
> > > > For example, there are still some spelling and grammar errors (e.g. Ada-mgering in L443), Figure 4 (a) is still hard to read and all of the citations seem to be formatted using \citet but many should be \citep.
> > > >
> > > > I will raise my score accordingly.

---

> > > > > ### Author Response · Authors · 2024-12-02
> > > > > **Thank You for Your Appreciation**
> > > > >
> > > > > Thank you very much for your appreciation! Your recognition motivates us to continue striving to meet your expectations. We are grateful for your constructive suggestions and will carefully address the issues you’ve highlighted in the revised manuscript. Thank you once again for your time and thoughtful input.
> > > > >
> > > > > Sincerely,
> > > > > The Authors

---

### Official Review · Reviewer_6MqJ · 2024-11-04

**Soundness:** 3
**Presentation:** 2
**Contribution:** 3
**Rating:** 8
**Confidence:** 3

**Summary:**

When merging models (generally finetuned on different tasks), many techniques boil down to a weighted sum (generally of "task vectors", the difference between the finetuned model and the pre-trained model) include _per-model_ scaling parameters. This creates an exponential number of settings and makes it intractable to try all the different possible merges.

Normally, merging methods are evaluated based on their average performance across many tasks, but they point out that this setting ignores the idea that a user may care more about performance on some subset of tasks than others. To capture this, they introduce the metric of the "win rate" how often a model on one method's Pareto frontier outperforms the models on another methods frontier.

They find that by sampling several _per-model_ scaling hyperparameters, they can use a quadratic approximation to create a better Pareto frontier with less computational resources.

**Strengths:**

The paper does a good job explaining the motivating the idea of using a Pareto frontier when evaluating model merging and a good job explaining their win-rate metric.

The paper gives a good overview of the quadratic approximation of the Pareto frontier.

**Weaknesses:**

The Pareto frontier based metric they use (win rate) is explained well, but during the comparisons to other common merging methods, it would have been nice to see another experiment that used their approach to set merging hyperparameters for those methods to see if greater average performance could be achieved. For example, comparing the avg performance of TIES with the hyperparameters from the original paper vs parameters found by their method.

Often merging methods are also evaluated on if they retain the ability to generalize to new tasks, it would be nice to see some experiments to test the generalization abilities of models merged with hyperparameters found using their method.

They include some talk of using Bayesian optimization for the sampling of hyperparameters and of using nested model merging, but their discussion (intro, methods, results, etc.) for these are so sparse they should probably be cut.

MAP is already a very common acronym for Maximum a Posteriori estimation. This collision will hurt adoption of their approach and is distracting as you need to keep reminding yourself is something else when you see MAP in their paper.

Figure 5 is designed to demonstrate the exponential growth of having _per-model_ hyperparameters. This growth is explained well enough in the paper that such a large figure is not the most effective use of space.

Nit: Lots of places where references appear to be part of the text, where they shouldn't be, i.e., it is Author (year) instead of (Author, year).

**Questions:**

N/A

---

> ### Author Response · Authors · 2024-11-20
>
> Thank you for the valuable comments and reviews! Below are our answers to the questions.
>
> ## Hyperparameters of the algorithms
>
> Q1: The Pareto frontier-based metric they use (win rate) is explained well, but during the comparisons to other common merging methods, it would have been nice to see another experiment that used their approach to set merging hyperparameters for those methods to see if greater average performance could be achieved. For example, comparing the avg performance of TIES with the hyperparameters from the original paper vs. parameters found by their method.
>
> A1: Thank you for the advice.
>
> As for the propose of combining TIES with MAP, we would like to point out that unlike TA and DARE, the scaling parameter for TIES is for the merged model as a whole instead of for individual task vectors and hence cannot be used for controlling the preference over tasks. We searched for hyperparameters of "k" and $\lambda$. They are the hyperparameters that need to be optimized, rather than some controls that could lead to trade-off performance on different tasks.
> We tried different combinations of "k" and $\lambda$ (searching step = 0.1). We end up using $\lambda=1$ and $k=20$ recommended by the original paper (P22, C.4) because this combination gives the best results and dominates other combinations. We would also like to point out that TIES-merging is very sensitive to the hyper-parameters. If we don't use this combination, their performance collapse quickly and dominated by both MAP and NMMAP.
> Specifically, in Algorithm 1 in TIES-merging (https://arxiv.org/pdf/2306.01708), $\tau_t$ is the task vector for task $t$, they obtain $\tau_{m}^{p}=\frac{1}{|\mathcal{A}^{p}|}\sum_{t\in\mathcal{A}^{p}}\hat{\tau}_{t}^{p}$ for $p$ in $1,\ldots, d$ and obtain the merged model as $$\theta_m\leftarrow\theta _\mathrm{init}+\lambda * \tau_m.$$ Note that here $\lambda$ is a scaling factor for the merged task vectors as a whole, and its value needs hyperparameter search, and the original TIES-merging paper recommends to use $\lambda=1$. ($\tau_m$ cannot be separated to a sum of different task vector from different tasks.)
> Unlike in task arithmetic where the merged model is $$\theta _m\leftarrow\theta _\mathrm{init}+ \sum _{t=1}^n \lambda_t * \tau_t$$ where the users can control the preference of task $t$ by setting $\lambda_t$, the $\lambda$ in TIES-merging is not task specific.
>
> ## Generalization on unseen tasks
>
> Q2: Often, merging methods are also evaluated on whether they retain the ability to generalize to new tasks. It would be nice to see some experiments to test the generalization abilities of models merged with hyperparameters found using their method.
>
> A2: We believe testing generalization on unseen tasks is an excellent approach. A suitable method for this is to identify tasks that share similarities with the training tasks. For instance, a multi-class classification task involving single images can encompass both cars and traffic signs. Our preliminary view is to use Cityscapes for the multi-class classification. Due to the time restriction, we will reserve this for future work.
>
> ## Figure 5 occupies space
>
> Q3: Figure 5 is designed to demonstrate the exponential growth of having per-model hyperparameters. This growth is explained well enough in the paper that such a large figure is not the most effective use of space.
>
> A3: Thanks for pointing this out. This figure is also designed to show the meaning of “pts_per_dim” since we didn’t find a good name for this concept. I think we could move this figure to the appendix as well, just in case people feel confused about the concept of “pts_per_dim.” We changed it in the updated manuscript.
>
> ## Discussion about Bayesian MAP and Nested MAP
>
> Q4: They include some talk of using Bayesian optimization for the sampling of hyperparameters and of using nested model merging, but their discussion (intro, methods, results, etc.) for these is so sparse that it should probably be cut.
>
> A4: We are sorry that, because of the space limit, we moved most discussions of Bayesian MAP (BMAP) and Nested MAP (NMAP) into the appendix. However, since we moved the original figure 5 to the appendix, we included more explanation of BMAP and NMAP in section 3.3.
>
> ## Name of the algorithm
>
> Q5: MAP is already a very common acronym for Maximum a Posteriori estimation. This collision will hurt the adoption of their approach and is distracting as you need to keep reminding yourself it is something else when you see MAP in their paper.
>
> A5: That’s a good point! We should rename it for sure. We think maybe LocMAP could be the new name. We changed it in the updated manuscript.
>
> ## Reference issue
>
> Q6: Lots of places where references appear to be part of the text, where they shouldn't be, i.e., it is Author (year) instead of (Author, year).
>
> A6: Thanks for pointing it out! Could you please kindly refer us to the line of one example? We couldn’t find it. Thank you!

---

> > ### Author Response · Authors · 2024-11-29
> > **Follow-up on Rebuttal and Review Feedback**
> >
> > Dear reviewer 6MqJ,
> >
> > We sincerely appreciate the time and efforts you've devoted to reviewing our work. We understand that your schedule may be quite busy. As the authors-reviewer discussion period draws to a close, we kindly request your attention to our responses. Our aim is to gain insights into whether our responses effectively address your concerns and to ascertain if there are any additional questions or points you would like to discuss. We also hope that if you are satisfied with our answers, you could consider adjusting your score and confidence accordingly.
> >
> > We look forward to the opportunity for further discussion with you. Thank you again very much for your thoughtful consideration.
> >
> > Best regards,
> > The Authors

---

> > > ### Author Response · Authors · 2024-12-02
> > >
> > > Dear reviewer 6MqJ,
> > >
> > > We hope this message finds you well. We wanted to kindly remind you that the response to our rebuttal is due today. Your feedback is very valuable to us, and we sincerely appreciate the effort you’ve dedicated to reviewing our work.
> > >
> > > If there’s anything we can clarify or support you with, please don’t hesitate to let us know. Thank you again for your time and thoughtful input!
> > >
> > > Best regards,
> > >
> > > The Authors

---

### Author Response · Authors · 2024-11-25
**TL;DR: Common rebuttal**

We sincerely thank all reviewers for their valuable recommendations and insightful advice. We have incorporated the suggested revisions into the manuscript and addressed all questions. Below, we summarize our responses:

---

### General Revisions
- We expanded the discussion on Bayesian MAP and Nested MAP in the main body, leveraging the space saved by moving Figure 5 to the appendix.
- We added more related references and discussed their relevance to our MAP approach in the newly updated **Discussion on Related Works** section.

---

### Responses to Reviewer 6MqJ
- We clarified why TIES-merging cannot be combined with MAP, highlighting that the scaling coefficient $\lambda$ is not task-specific.
- We proposed potential methods for evaluating generalization across unseen but related tasks.
- As suggested, we moved Figure 5 to the appendix, as the concept was deemed clear but the figure occupied significant space in the main text.
- Following the recommendation, we renamed the algorithm from "MAP" to "LocMAP" to avoid confusion, as MAP is widely understood to mean "Maximum a Posteriori."

---

### Responses to Reviewer 97hV
- We relocated the motivation section to Section 3 (after introducing the algorithm) to improve clarity and avoid confusion.
- We elaborated on the nonlinear warper and provided additional explanation of the definitions for direct search and its grids.

---

### Additional Updates
- We compared model-merging-based methods with gradient-descent-based methods, emphasizing the advantages of the former: computational efficiency, avoiding retraining for different preference weightings, and better handling of task trade-offs.
- We addressed the reviewer’s suggested works, **Knowledge Fusion by Evolving Weight** and **It’s Morphing Time (MM-MO)**, and discussed their correlation with our approach.
- We conducted new experiments **comparing MAP with established MOOP algorithms, MOEA/D**, and included a detailed analysis of its performance.
- We added experiments **evaluating MAP on LLM tasks, specifically merging Math LLM with Coding LLM**, demonstrating its efficacy in this context.
- Additional experiments **compared MAP with recent baselines such as Adamerging++ and DELLA-Merging**, showing MAP’s superiority in preference-weighted accuracy.

**Please note that all new experiments and updates have been incorporated into the revised manuscript.**

---

### Meta-Review · Area_Chair_G8wd · 2024-12-14

**Metareview:**

The paper looks at model merging for creating a model to perform on known tasks.
Current merging methods often propose a single merged models, this paper proposes a new method to create multiple models that have different capabilities as shown by tradeoffs between which evaluated dataset gets better scores.

Strengths:
New algorithm.
Clear claims.

Weaknesses:
Presentation.

**Additional Comments On Reviewer Discussion:**

Despite private and public urges, there was none... (except 97hV that acknowledged the presentation issues)
Some suggested weaknesses were addressed in the rebuttal (even if no discussion appeared).

---

### Decision · Program_Chairs · 2025-01-22

Accept (Poster)